# The radiation continuum and the evolution of frog diversity

Gen Morinaga [1,2], John J. Wiens [3] & Daniel S. Moen [1,4]

Most of life's vast diversity of species and phenotypes is often attributed to adaptive radiation. Yet its contribution to species and phenotypic diversity of a major group has not been examined. Two key questions remain unresolved. First, what proportion of clades show macroevolutionary dynamics similar to adaptive radiations? Second, what proportion of overall species richness and phenotypic diversity do these adaptive-radiation-like clades contain? We address these questions with phylogenetic and morphological data for 1226 frog species across 43 families (which represent >99% of all species). Less than half of frog families resembled adaptive radiations (with rapid diversification and morphological evolution). Yet, these adaptive-radiation-like clades encompassed ~75% of both morphological and species diversity, despite rapid rates in other clades (e.g., non-adaptive radiations). Overall, we support the importance of adaptive-radiation-like evolution for explaining diversity patterns and provide a framework for characterizing macroevolutionary dynamics and diversity patterns in other groups.

Adaptive radiation is characterized by high rates of species diversification and phenotypic evolution[1–3]. Over time, such high rates should produce more species and greater phenotypic diversity than clades of similar age that have lower rates. Thus, adaptive radiation has often been ascribed a central role in generating life's diversity[2,4,5]. However, some authors have questioned whether diversification rates explain variation in species richness among clades[6,7]. Furthermore, several studies have also found decoupled rates of diversification and morphological evolution[1,8–11], and others show that high rates of phenotypic evolution need not lead to high phenotypic diversity[12–14]. Overall, the assumption that adaptive radiation explains most of life's diversity remains untested.

Many other types of evolutionary dynamics may explain this diversity instead. For example, non-adaptive radiations[15,16] are characterized by high rates of diversification but limited phenotypic evolution. All else being equal, these clades may encompass as much species diversity as adaptive radiations[17,18], but with limited phenotypic diversity. Similarly, clades with high rates of phenotypic evolution but low diversification rates ("adaptive non-radiations")[1] may produce as much phenotypic diversity as adaptive radiations but limited species

diversity. Yet, to our knowledge, the phenotypic and species diversity produced by these different evolutionary dynamics has not been quantified, nor has a framework been developed to do so.

Here, we develop such a framework and apply it to anuran amphibians. Anurans (frogs and toads, "frogs" hereafter) are an excellent group for studying the origins of species richness and phenotypic diversity. Anura includes >7400 extant, described species[19]. This diversity is distributed across 54 families that vary broadly in species richness, from 1 to >1000 species[19]. Anurans also show eco-morphological and physiological specializations for different micro-habitats (e.g., aquatic, arboreal, terrestrial)[20–26]. Families vary in the number of species using each microhabitat, with some families containing multiple types of microhabitat specialists and others containing only one[23]. Importantly, diversification rates are correlated with species richness across families[23]. However, it is unclear whether species-rich clades with high diversification rates are also morphologically diverse. Thus, three key questions remain unanswered: (i) How do rates of morphological evolution relate to overall morphological diversity? (ii) Are rates of diversification and morphological evolution correlated? (iii) To what extent do clades that rapidly diversified and

[1]Department of Integrative Biology, Oklahoma State University, Stillwater, OK 74078, USA. [2]Faculty of Veterinary Medicine, University of Calgary, Calgary, AB T2N 4N1, Canada. [3]Department of Ecology and Evolutionary Biology, University of Arizona, Tucson, AZ 85721, USA. [4]Department of Evolution, Ecology, and Organismal Biology, University of California, Riverside, CA 92521, USA. ✉e-mail: dmoen@ucr.edu

rapidly evolved morphologically explain overall anuran richness and phenotypic diversity?

Studies of phenotypic diversity and evolution are often limited by a lack of large-scale multivariate morphological datasets. Here, we generated such a dataset using 10 ecologically relevant morphological traits. We then estimated rates of multivariate evolution. We analyzed the relationship between morphological rates and morphological diversity, and between species diversity and net diversification rates. We used the two types of rates to characterize a two-dimensional radiation space for frogs. We show that all four major types of macroevolutionary outcomes are represented: (i) high rates of diversification and phenotypic evolution, (ii) low rates of diversification and phenotypic change, (iii) high phenotypic rates but low diversification rates, and (iv) high diversification rates with low phenotypic rates. Most importantly, we show that families that have adaptive-radiation-like evolution (high diversification and phenotypic rates) contain the majority of species diversity and phenotypic diversity across a major clade.

## Results

### The anuran morphospace

We first characterized morphological diversity across frogs. We measured 4628 adult specimens of 1234 anuran species from around the world, including 51 of 54 families[19] and on average 25% of the species in each family (range = 6.75–100%; $r = 0.923$ between sampled and described richness of families). We quantified body shape using 10 ecologically relevant traits[21,22,27,28]. We also included microhabitat[23] and phylogenetic data[29]. We size-corrected our raw morphological data[30–32] and summarized diversity in body form using phylogenetic principal components analysis[33].

The first five phylogenetic principal components axes (pPC1–pPC5) collectively explained 92% of the variation among sampled species (Fig. 1; Supplementary Table 1). These five axes characterized variation in shape that distinguished the major frog ecomorphs that are each associated with a different microhabitat (Supplementary Table 1 and Supplementary Fig. 1). Considering this morphological variation along with the species richness of each family reveals that some families have high species richness and occupy a broad morphospace (Fig. 1a), whereas others have high richness and a narrow morphospace (Fig. 1b), low richness and a broad morphospace (Fig. 1c), or low richness and a narrow morphospace (Fig. 1d).

### Relationship between rates and diversity

We next examined how rates of species diversification and morphological evolution were related to each other and to their respective types of diversity. We defined morphological diversity as the volume of $n$-dimensional morphospace occupied by a set of species, similar to that of Hutchinson[34] for niches. We quantified this volume using two approaches: a convex-hull volume[35] and a hypervolume[36]. Morphological diversity was summarized using the first five pPCA axes. Convex-hull volumes and hypervolumes were strongly, positively correlated ($r = 0.977$; $P < 0.001$) across the 27 families for which volumes could be calculated. Given this similarity, we focus on hypervolume given its increased biological realism (see Methods).

We calculated multivariate rates of morphological evolution[37] using size-corrected species means. In absolute rates (i.e., calculated on logged data but before centering and scaling them for downstream analyses), families varied >50-fold, from 0.00010–0.00593 (unitless after size-standardization), with mean=0.00131 ($n = 43$). Rates of morphological evolution and morphological diversity were significantly correlated among families ($r = 0.633$; $P < 0.001$; $n = 27$; Fig. 2), but considerable phenotypic diversity remained unexplained by variation in rates.

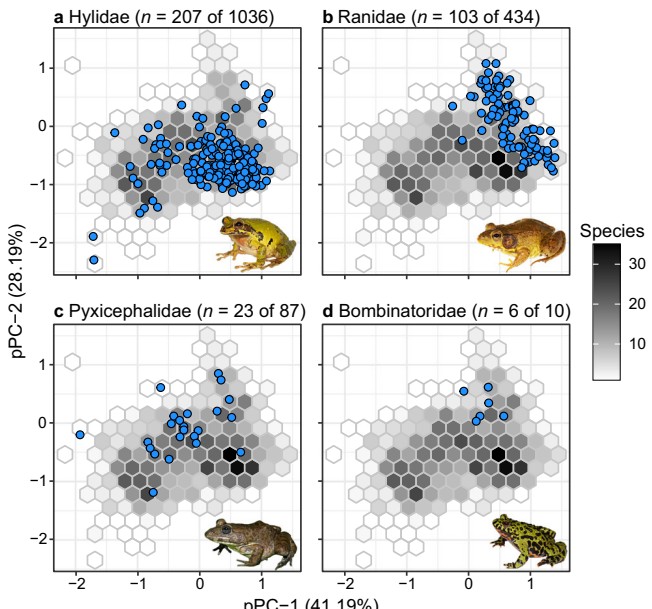

**Fig. 1 | Morphological and species diversity across representative families in our dataset.** Plots of the first two phylogenetic principal component axes demonstrate how morphological and species diversity may be somewhat uncoupled across families: **a** high in both; **b** high species diversity, narrow morphospace; **c** low species diversity, broad morphospace; **d** low in both. *n* indicates our sampling relative to the total species diversity of each family (latter number). Blue dots represent individual species. Grey hexagons show density of all of species in our dataset. Photos represent each family: **a** *Smilisca baudinii*, **b** *Rana clamitans*, **c** *Aubria subsigillata*, and **d** *Bombina orientalis*. Photos in **a**, **b**, and **d** by D.S.M.; photo in **c** used with permission by Daniel Portik. Source Data can be found within Supplementary Code 1.

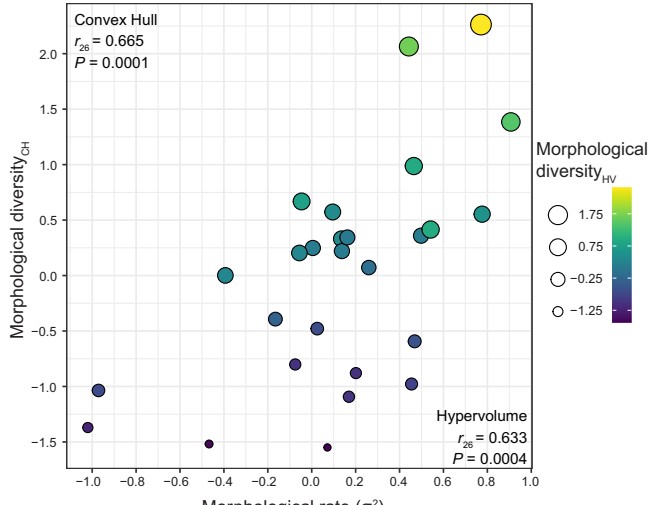

**Fig. 2 | Morphological diversity versus rate of evolution.** Diversity is represented by five-dimensional morphological volumes defined by pPC1–5 using convex hull (CH) and hypervolume (HV) methods, of which we took the 5th-root, scaled, and mean-centered. Each dot represents a family with more than six species sampled in our morphological dataset ($n = 27$), with convex-hull volume on the vertical axis and hypervolume indicated by dot area and color. Phylogenetic generalized least-squares correlation results between morphological rate and each method of volume estimation are also shown. Both *P*-values reflect two-sided hypothesis tests with no adjustment for multiple comparisons. Source Data can be found within Supplementary Code 1.

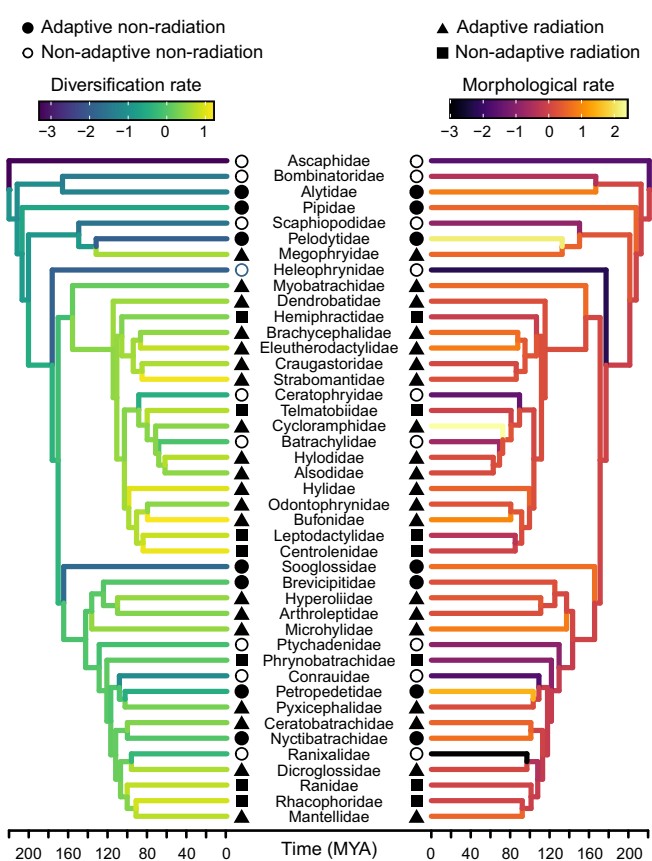

**Fig. 3 | Net diversification rates and rates of morphological evolution of each anuran family included in our analysis ($n = 43$).** There are 11 other families of anurans that together include <1% of anuran species. These 11 families had too few species to estimate their rates of morphological evolution. Net diversification rates plotted here assumed a moderate extinction fraction ($\varepsilon = 0.5$). Both rates were ln-transformed, mean-centered, and scaled by their standard deviations. Symbols indicate to which quadrant of the radiation space each family was assigned based on their combined rates of diversification and morphological evolution (see Fig. 4). Source Data can be found within Supplementary Code 1.

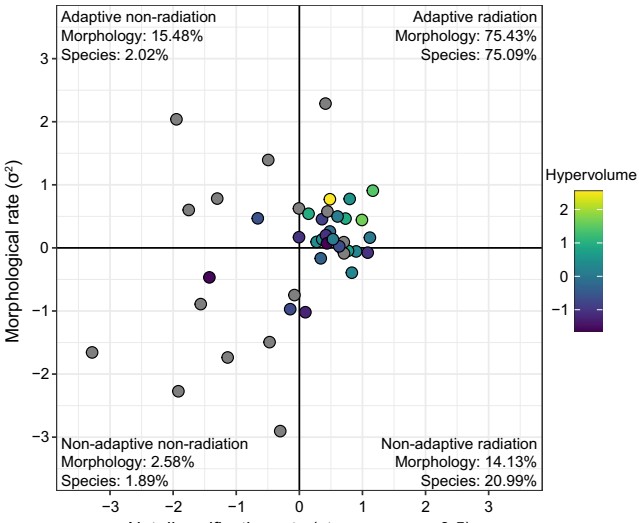

**Fig. 4 | The radiation space of frogs.** Dots show rates of net diversification and multivariate morphological evolution for frog families ($n = 43$). Rates are logged, mean-centered, and scaled. Dot color indicates morphological diversity defined by a five-dimensional hypervolume, with gray dots representing families whose volume could not be estimated given too few species. Percentages indicate the species diversity and morphological diversity that each quadrant represents. Morphological percentages across quadrants may sum to >100% when quadrant morphospaces overlap or <100% if gaps occur between quadrant morphospaces. Source Data can be found within Supplementary Code 1.

We next used species richness[19] and family ages[29] to calculate net diversification rates of families, using method-of-moments estimators[38]. We found strong correlations between richness and diversification rates among sampled families ($r = 0.891$; $P < 0.001$; $n = 43$). In contrast, rates of diversification and morphological evolution were more weakly correlated across families ($r = 0.355$; $P = 0.019$; $n = 43$; Fig. 3). Results were highly similar using birth-death estimators that utilize within-family branch lengths (Supplementary Table 2).

**Diversity and the radiation continuum**

We formalize here the idea of a "radiation space", where clades fall on a spectrum of variation from low rates of diversification and morphological evolution to high rates in both[39]. The correlation (or lack thereof) of rates of net diversification and morphological evolution defines this space. In principle, clades could fall along a linear continuum if this rate correlation were strong, with clades distributed continuously from low rates in both to high rates in both. Alternatively, low correlation or no correlation would instead suggest that the continuum is better represented by a two-dimensional space, where some clades show high rates of one type with low rates of the other. Because we found net diversification rates and rates of morphological evolution were only weakly correlated, we considered a two-dimensional radiation space.

We then divided this space into quadrants of adaptive radiation (high rates of both diversification and morphological evolution), non-adaptive radiation (low morphological, high diversification), adaptive non-radiation (high morphological, low diversification), and non-adaptive non-radiation (low rates of both)[1,3,15,17,18]. We followed previous authors who defined adaptive radiation as a combination of high rates of both species diversification and phenotypic evolution[1–3,40]. However, limiting the term "adaptive radiation" only to clades that show significantly elevated rates would mean that almost all clades have unexceptional rates of diversification and morphological evolution (even those with very high rates). Therefore, we instead considered the quadrants to indicate evolution that has been more like one type than another (e.g., "adaptive-radiation-like" clades, rather than "adaptive radiation" in the strict sense). The exact boundaries of these quadrants can be defined in various ways; here we used mean and median rates to delimit quadrant boundaries. However, the limits and range of the rates that characterize these boundaries must be group-specific: rates of diversification and phenotypic evolution can vary by orders of magnitude among clades[41–44]. No concept of adaptive radiation only ascribes the phenomenon to a single clade with the highest rates (e.g., plants).

We primarily used mean values of rates to define quadrants (see next section for alternative characterizations). We found that quadrant location of families was more-or-less evenly distributed across the phylogeny of families (Fig. 3). Furthermore, analyses of phylogenetic clustering using the $D$ statistic[45] showed neither significant clustering nor overdispersion across the tree for any type of evolutionary dynamic (Supplementary Table 3).

Among the 43 families included, 20 (46.5%) had above-average rates of both net diversification and morphological evolution (Fig. 4; Supplementary Fig. 2). Therefore, they showed evolutionary dynamics more consistent with adaptive radiation. Among these 20 families, four are notably species-rich (i.e., >400 species; Bufonidae, Hylidae, Microhylidae, and Strabomantidae) and include families that span frog morphospace (e.g., Hylidae in Fig. 1a). These families collectively

accounted for 75.1% of anuran species diversity and occupied 75.4% of anuran morphospace. While such clades explained the majority of species and morphological diversity across Anura, they clearly did not explain nearly all diversity, as is often posited[2,4,5].

By contrast, nine families had low rates for both diversification and morphological evolution (non-adaptive, non-radiation quadrant; Fig. 4). Four of these families were exceptionally species-poor with <10 species each (Ascaphidae, Conrauidae, Heleophrynidae, and Scaphiopodidae; Supplementary Fig. 2). As expected, these nine families represented very small percentages of anuran diversity, including 1.9% of species diversity and 2.6% of the total morphospace.

Seven families had high morphological rates but low diversification rates (adaptive non-radiation quadrant; Fig. 4). Consistent with these rates, these families accounted for more anuran morphospace (15.5%) than species diversity (2.0%). Another seven families showed the opposite pattern, with high net diversification rates but low morphological rates (non-adaptive radiation quadrant; Fig. 4). Among these families, Rhacophoridae and Ranidae had high species richness but limited morphological diversity (Fig. 1b). As expected, this quadrant included more of anuran species diversity (21.0%) than morphological diversity (14.1%). More surprisingly, the morphospace occupation of this non-adaptive radiation quadrant was similar to that of families in the adaptive non-radiation quadrant (i.e., with high rates of morphological evolution).

Why might non-adaptive radiations (i.e., low rates of phenotypic evolution) explain similar morphological diversity as adaptive non-radiations (i.e., high rates of phenotypic evolution)? Given a positive correlation between rates of morphological evolution and morphological diversity (Fig. 2), we do not expect non-adaptive radiations to accumulate much morphological diversity over long periods of time.

We see two possible explanations. First, although the correlation between morphological rates and morphological diversity is relatively high (Fig. 2), the key exceptions to this general trend (e.g., low-rate, high-diversity clades) may contain much morphological diversity. This could happen if a clade with a low rate of morphological evolution generates more morphological diversity simply by having more species or being older: a low rate, realized over many (or long) evolutionary branches, could decouple phenotypic rates and diversity[46]. We see this potentially manifested in Batrachylidae, Phrynobatrachidae, and Ptychadenidae (Supplementary Fig. 2), of which Batrachylidae is the youngest and has the fewest species. Second, some clades with low rates of morphological evolution may represent distinct parts of anuran morphospace. These clades may collectively cover a large proportion of anuran morphospace without having high rates of morphological evolution within families. For example, the non-adaptive radiation quadrant includes clades that are mostly aquatic (Ranidae, Telmatobiidae), arboreal (Centrolenidae, Hemiphractidae, and Rhacophoridae), and terrestrial (Leptodactylidae). These ecomorphs collectively span anuran morphospace (Supplementary Fig. 1). In contrast, rapidly evolving clades in the adaptive non-radiation quadrant do not show such ecomorphological diversity (e.g., no predominantly arboreal clades)[23], which may have led to somewhat less morphological diversity in this quadrant than expected.

These patterns illustrate how explaining species richness and morphological diversity are complementary but not identical goals. When comparing groups of similar age, high net diversification rates necessarily produce more species than low rates. In contrast, high morphological rates can lead to high diversity, but they can also produce repeated instances of the same morphotypes, thus contributing little to overall morphological diversity. In anurans, this might mean repeated evolution of the same convergent ecomorphs[22], driving up rates of evolution without expanding overall anuran morphospace. Alternatively, a distinct morphotype may evolve in the common ancestor of a clade and extend overall anuran morphospace, even if

rates of morphological evolution within that clade are low (e.g., the giant-headed Ceratophryidae)[47,48].

## Alternative characterizations of clades and diversification rates

Our results were generally robust to alternative methodological choices. Calculating net diversification rates with birth-death estimators produced very similar results, with the only notable difference being a 7% shift in morphological diversity from adaptive non-radiation to the adaptive-radiation quadrant (Supplementary Fig. 2). We found similar qualitative results when we used crown ages and different extinction fractions (Supplementary Fig. 3). However, when using crown ages, adaptive non-radiations consistently contained a greater proportion of morphological and species diversity than when using stem ages. Using median rates to delineate quadrant bounds (instead of means) slightly reduced the proportions of species and phenotypic diversity attributed to adaptive radiation (Supplementary Fig. 2).

We found the same general patterns when analyzing clades delimited at 80, 100, and 120 million-year intervals instead of using families (Supplementary Fig. 4). Specifically, the adaptive-radiation quadrant still included most species diversity and morphological diversity. However, when clades were defined using increasingly older clade ages, adaptive-radiation-like clades explained increasingly more diversity. When clades were delimited at 120 million years old, the adaptive-radiation-like clades contained up to 98% of species diversity and 96% of morphological diversity (Supplementary Fig. 4). This pattern contrasts with the family-level analyses (~75% for both) and occurs because 120 million-year-old clades with high species diversity also contain most anuran morphological diversity (e.g., Hyloidea includes 20 families, 53% of anuran species, and all ecomorph types)[23]. Future studies should test whether this pattern (i.e., adaptive-radiation-like clades explain more diversity when clades are older) applies broadly to other organisms.

## Discussion

Adaptive radiation has become a central topic in evolutionary biology likely because of the untested idea that adaptive radiations are responsible for much of life's species diversity and phenotypic diversity[3]. We show here that in frogs, clades that are rapidly evolving and diversifying (i.e., adaptive-radiation-like) contain ~75% of both phenotypic and species diversity. The high diversity in this quadrant occurs in part because these clades have high rates of diversification and phenotypic evolution, but also because nearly half of anuran families are in this quadrant. We also show that the rest of phenotypic and species diversity is distributed among clades that show various combinations of fast and slow rates of phenotypic evolution and net diversification. However, we acknowledge that our results are for one group (frogs), and not all of life.

Could these patterns be more general? Given that our radiation-space framework has not yet been applied to other groups, it is unclear how rates of diversification and phenotypic evolution will explain diversity in other taxa. However, variation in diversification rates (alone) explains much variation in species diversity among clades of the same rank[41] (e.g., families, phyla, kingdoms) across life, and within major groups (e.g., plants, animals, fungi). Some studies have also identified the correlates of this variation in diversification rates[49–51]. While these correlates are often morphological (e.g., multicellularity)[52], they may instead involve non-morphological factors like range size and climatic niches[50,51,53].

Nevertheless, explaining species richness is only one goal. Explaining phenotypic diversity remains more challenging. Few studies have compared rates of multivariate phenotypic evolution to overall clade-level phenotypic diversity[12,14,54]. We found that phenotypic diversity was correlated with rates of phenotypic evolution among clades ($r = 0.63$; Fig. 2). Thus, these rates explain considerable variation in phenotypic diversity among clades. Yet why do some clades show

higher rates of morphological evolution than others? High rates of multivariate morphological evolution may be explained by high microhabitat diversity, since different microhabitats are associated with different body shapes[22,26,27,55]. Furthermore, ecomorph and body-size evolution may be accelerated after colonization of regions with few competing lineages[1,56,57]. We found partial support for these expectations. Families with the lowest rates (Ascaphidae, Conrauidae, Heleophrynidae, Ranixalidae; Supplementary Fig. 2) do have small ranges and co-occur with clades with complementary ecological roles[23,58]. Yet some families with the highest rates show modest microhabitat diversity and very different range sizes (Bufonidae is globally distributed, whereas Cycloramphidae is restricted to southeastern Brazil)[19]. Surprisingly, other high-rate families (Pelodytidae, Petropedetidae) have low microhabitat diversity[23] and limited geographic ranges[19]. Future studies should simultaneously analyze multiple factors, including ecological diversity, biogeography, competition, and life history[59].

We appreciate that readers may potentially have methodological concerns about our study, which we address here. The most important may be that we define adaptive radiation as high rates of net diversification and phenotypic evolution, and high rates often lead to high diversity. Given this definition, was it therefore inevitable that adaptive-radiation-like clades explained most species and phenotypic diversity? We see two reasons why not. First, the distribution of phenotypic and species diversity among quadrants also depends on the number of clades in each quadrant. For example, if we found only a few adaptive-radiation-like clades (instead of 20), most diversity might have been present instead among the adaptive non-radiations and non-adaptive radiations. The second reason is that even if clades were evenly distributed among quadrants (and even if rates determined most diversity), adaptively radiating clades need not contain more diversity than other clades. For example, all things being equal, adaptive radiations and non-adaptive radiations should have similar net diversification rates on average, which should lead to both types accounting for similar species diversity[17]. Moreover, adaptive non-radiations and adaptive radiations could account for similar phenotypic diversity, because they both have high phenotypic rates (and differ in their net diversification rates). Differences in diversification rates may lead to differences in species diversity, and in two clades with similar rates of phenotypic evolution but different numbers of species, the clade with higher species diversity is expected to have higher phenotypic diversity (all else being equal)[46,60]. Yet higher net diversification rates do not always lead to higher species diversity, as when young clades with high rates have fewer species than old clades with lower rates (e.g., plants vs. animals)[41].

A related methodological concern is that high diversification rates might inevitably be correlated with high rates of phenotypic evolution, since both rates depend on time. In fact, these two rates are only weakly related in anurans ($r = 0.355$), leading to 14 of 43 families (33%) falling in quadrants that show decoupled rates (i.e., high in one rate and low in the other). Further analysis shows that rates of morphological evolution are only weakly related to time per se (Supplementary Fig. 5), which could explain the low correlation with net diversification rates (see final section of Methods). Clades could also have high species diversity and phenotypic diversity because diversity accumulates steadily over time in older clades, meaning diversity is decoupled from rates. But this is not the pattern that we see.

We recognize that the exact quadrant boundaries used here are specific to our data. No definition of adaptive radiation is based on specific rate values[3,5,61,62], presumably because rates vary dramatically across organisms[41,42,44]. Thus, any similar approach for understanding the macroevolutionary drivers of diversity patterns among clades should be based on the observed data.

A related concern is that defining adaptive radiations is controversial[4,5]. We considered adaptive radiations to be clades with accelerated rates of diversification and ecologically relevant phenotypic

evolution, consistent with the widespread definition used by Schluter[3] and many others. Moreover, this definition focuses on linking phenotypes and environments, as well as showing utility of different phenotypes via studies of their performance[3]. Previous work has shown links between the morphological variables measured here and microhabitat use[22], diet[63], and locomotor performance[1,21,26,55]. However, the phenotypic diversity quantified here only relates to the traits we measured. Other variables might be relevant to frog diversification and phenotypic evolution, such as calls, reproductive modes, and coloration. Yet these variables generally show no relationship with diversification rates in anurans[64–66] (but see ref. [67]). In a similar vein, we have not included all frog species when estimating phenotypic diversity, but we sampled all major groups and ecomorphs. Thus our sample should be representative of the major patterns in anuran species and phenotypic diversity.

Methods for estimating diversification rates are also contentious. Yet, the two methods used here utilize different information (i.e., species diversity and age versus branch-length distributions) but yielded highly correlated rates ($r = 0.74$–$0.78$; Supplementary Table 2). Moreover, the primary method we used is demonstrably accurate in simulations, including simulations with variable rates over time[68–70]. It also accounts for extinction via an extinction fraction. Estimating extinction rates without fossil data is controversial[71–73], and the fossil record is poor for most anuran families[74,75]. Thus, we considered various possible extinction fractions and found our results were largely insensitive to fractions ranging from no extinction ($\varepsilon = 0$) to very high ($\varepsilon = 0.9$) extinction (Supplementary Fig. 3). Furthermore, we found very similar results when using an alternative diversification method (birth-death estimator) that estimates extinction rates from species-level phylogenies within families (Supplementary Fig. 2). Overall, we emphasize that our study does not assume uniform diversification dynamics over time and that our results are robust to alternative diversification-rate estimators.

In summary, adaptive radiations (clades with high rates of diversification and phenotypic evolution) are thought to contain much of the phenotypic and species diversity of life. But how much exactly? Here, we have quantified this percentage in a major clade, showing that for frogs the answer is about 75%. That is, those 20 clades (~45%) with higher rates of phenotypic evolution and species diversification (adaptive-radiation-like) disproportionally represent the morphological diversity and species richness of frogs. This occurs even though phenotypic and diversification rates are only weakly correlated with each other among clades. We also find that other frog clades show various combinations of low and high rates of phenotypic evolution and diversification. These other types of clades (not adaptive-radiation-like) include >50% of sampled clades and ~25% of extant species and morphological diversity. Beyond anurans, we provide a framework for addressing this question that can be readily applied to other groups of organisms. We expect that clades with high diversification rates will contain much of the species richness in other major groups, but the contribution of fast-evolving clades to overall phenotypic diversity in other groups remains to be seen. The limiting factor for future analyses of this type may be high-quality morphological data for large numbers of species, which will allow a group's overall morphological diversity to be estimated.

## Methods

This study contained no experimental component or data collection on live animals and so no ethical oversight was necessary.

### Morphological data collection

We measured 4628 adult museum specimens of 1234 species from around the world. Most of these data were novel, whereas 901 specimens from 194 species came from previously published datasets[1,20–22]. Our sample included 51 of 54 anuran families[19]. The three remaining families (Calyptocephalellidae, Ceuthomantidae, and Nasikabatrachidae) are scarce in museum collections. We chose species within

families based on their availability in museum collections, with species sampling proportional to the described species diversity of each family ($r = 0.923$). However, for eight families we were only able to sample a single species, which prevented calculating rates of morphological evolution. Thus, we excluded them (to total 1226 species from 43 families) from those analyses and all downstream analyses based on those rates. We also note that some studies of rates of morphological evolution have removed clades with low numbers of species (e.g., less than four[8]). In our dataset, 11 families had between 2–4 species sampled for morphological data. However, some of these families have four or fewer total extant species, and thus excluding these families would result in biasing our analyses to ignore clades with low species richness. Moreover, while lower sampling may increase the variance in estimates of a clade's true rate of evolution, such estimates are unbiased[1]. Finally, to reduce potential effects of sexual-size dimorphism on our sampling[76–78], we measured male specimens when possible (89% of all specimens sampled; 82% of our sampled species were represented only by males). Males tend to be better represented in collections than females, presumably because of their calling behavior. We include all raw intraspecific data as Supplementary Data 1 and species means, sample sizes, standard deviations, and standard errors as Supplementary Data 2.

We quantified body shape using linear, area, and volumetric measurements of traits that are ecologically and functionally relevant to locomotion and microhabitat use[21,22,27,28]. First, we measured snout-vent length, head length, head width, upper arm, forearm, hand, thigh, crus, tarsus, and foot lengths to the nearest 0.01 mm using a Mitutoyo digital caliper (Kanagawa, Japan). We took each measurement only once, as our measurements were highly precise; preliminary repeated measurements showed a coefficient of variation of less than 0.03 for all measurements, with most <0.015. We summed the linear limb element measurements together (i.e., front limb length, hindlimb length). Second, we photographed the foot and hand of each specimen and measured the areas of digit tips on both the front and hind limb, interdigital webbing of the hind limb, and the inner metatarsal tubercle using ImageJ[79]. We summed the areas of the digit tips separately for the front and hind limbs and interdigital webbing across the foot. Detailed descriptions of all measurements are given in Supplementary Table 4.

Finally, we quantified leg muscle volume using external linear measurements. We used thigh and crus muscle volume among the traits for characterizing anuran body shape. Muscle mass is strongly related to locomotor performance and microhabitat use in anurans[21,22,26,55]. However, we could not calculate mass by dissecting muscle tissue from museum specimens at this scale of sampling. Thus, we estimated leg muscle volume, which should scale 1:1 with mass[80] and could be quantified using external linear measurements. We estimated muscle volume of the thigh and crus separately, considering each leg segment as two cones sharing an elliptical base (i.e., the approximate cross-sectional area of the underlying muscle). We measured the depth and width of the thigh and crus at their mid-points as the axes of the ellipse. To ensure our approximation of muscle volume adequately represented its mass, we took advantage of the previously published subset of our data (641 specimens from 132 species[21,22]) that included masses of dissected thigh and crus muscles. For these specimens, we natural-log transformed (ln) thigh and crus masses and volumes to linearize the relationship, then checked the correlation between thigh (or crus) muscle mass against estimated volume at the specimen level. We found that mass and volume were strongly correlated ($r = 0.974$ and $0.965$ for thigh and crus, respectively), which suggests that our volume approximation accurately represents muscle mass.

We lacked width and depth measurements but had muscle masses and lengths for thigh and crus for 238 specimens from 49 species. To include these 238 specimens in our analyses, we estimated the muscle cross-sectional area, which we could then use with observed leg segment length to estimate volume. We thus regressed the ln thigh (or crus) cross-sectional area on ln mass for the aforementioned 641 specimens with both data. We then used this model to predict cross-sectional areas for the 238 specimens that lacked width and depth measurements. These regressions showed that the mass of thigh and crus strongly predicted cross-sectional area ($R^2 = 0.949$ and $0.931$ for thigh and crus, respectively; Supplementary Fig. 6).

## Microhabitat states

We used previously published microhabitat data[23] and additional natural history descriptions to classify most species to microhabitats; new classifications determined for this study are provided in Supplementary Data 3. Most species can be categorized into eight different microhabitat states[22,23]. Four of these states are "base" microhabitat states that broadly categorize adult frog ecology: aquatic (found primarily in water), arboreal (found primarily in trees and brushes), burrowing (found primarily in self-dug burrows), and terrestrial (found primarily on the surface or under shallow leaf litter). Three additional categories combine terrestrial microhabitats with others, when ecological descriptions indicate that species spend time in both microhabitats: semi-aquatic, semi-arboreal, and semi-burrowing. The torrential microhabitat is characterized by occupying vegetation and rocks along high-gradient streams and rushing waters, thus combining aquatic and arboreal states.

## Phylogeny for comparative analyses

We used the posterior distribution of time-calibrated, multi-locus trees generated by Jetz and Pyron[29] for comparative analyses. We chose this phylogeny because it included all species in our morphological dataset. Whereas most more recent phylogenies[81,82] may have more molecular data per species and potentially more accurate clade ages, they have far fewer taxa (i.e., they would leave out about 90% of our species). Moreover, recent comparative analyses of diversification rates in anuran families show similar results regardless of the tree used to calculate clade ages[1].

We first pruned the posterior distribution to include only anuran species with genetic data (3449 species), because trees with taxa placed based on taxonomy alone may inflate rates of phenotypic evolution[83]. We used tools available at VertLife (www.vertlife.org/phylosubsets; date accessed: 25 January 2021) to download a random draw of 1000 trees. We then used TreeAnnotator[84] to calculate the maximum-clade credibility (MCC) topology and summarize branch lengths in millions of years, doing so with the "Common Ancestor heights" option. This option generally produces more accurate estimates of clade age than mean branch lengths[85].

## Size correction and visualization of morphology

Previous analyses have shown that adaptive morphological diversification in frogs is often unrelated to body size[1,21,22,86]. Thus, to focus on shape-based morphology, we size-corrected each trait using log-shape ratios[30–32], wherein we divided variables by SVL and then ln-transformed the resulting ratios[32]. Traditional log-shape ratios consider size as the geometric mean of all morphological variables[31]. However, we only used SVL as a metric of size, given that we measured the other variables precisely because we expected them to differ based on ecology[21,22]. By contrast, SVL does not differ based on microhabitat[22] and can differ greatly among species with similar body shape (e.g., refs. 57,63). For area and volume measurements, we took the square root or cube root of the raw values prior to size-correction to ensure equal scaling across variables[80]. We performed all size corrections on raw (i.e., intraspecific) data, then calculated species means from the size-corrected intraspecific values. For this and nearly all other analyses, we used the R computing environment[87], version 4.1.0.

To ensure that size standardization did not affect pPC axis interpretation, we also performed interspecific size-correction using

residuals[33] of each trait regressed against SVL, using *phytools* in R. We then conducted a phylogenetic PCA on these residuals. We found high correlations between the eigenvectors of each PC axis resulting from this alternative method of size standardization and our preferred ratio method ($r_{Mantel}$ = 0.987; $P < 0.001$). Thus, the method of size standardization is unlikely to change our interpretation of downstream analyses[30]. Furthermore, several papers have cautioned against treating residuals from linear regressions as data[88–90]. For these reasons and for brevity, we only present results obtained from the log-shape ratio method of size-correction.

We summarized diversity in body form using a phylogenetic principal components analysis (PCA) on species means, as implemented in the *phytools* package[91], version 0.7–47. We included size-corrected measurements described above of head length and width, front and hindlimb lengths, volumes of the thigh and crus muscles, areas of foot webbing and the inner metatarsal tubercle, and area of the digit tips of the foot and hand. We assumed a Brownian motion model of evolution, and we conducted the PCA on the phenotypic covariance matrix, given our prior standardization of all variables to the same scale and units[92]. We also performed a non-phylogenetic PCA to ensure that the interpretation of body form was insensitive to analytical method[92,93]. We compared the results of these two types of PCA by conducting a Mantel test (10,000 permutations) on the PCA eigenvectors, as implemented in the package *vegan*[94] version 2.5.7. This analysis showed a strong correlation ($r_{Mantel}$ = 0.885; $P < 0.001$) between phylogenetic and non-phylogenetic PCAs. Thus, PCA method seemed unlikely to affect downstream analyses or interpretations, so we used the resulting phylogenetic PCA scores for later analyses of morphological diversity.

## Units of statistical analysis of rates, diversity, and the radiation continuum

Our approach necessitated comparing many different clades. We chose families as the unit of analysis. Anuran families range from 1 to >1000 species and show substantial variation in diversification rates[23]. Families are also sufficient in number (54 total) to examine patterns with robust statistics. At shallower taxonomic levels (e.g., genera), we may see similar patterns as families[57] but would generally have too few species per clade to robustly calculate rates of phenotypic evolution. In contrast, anurans have relatively few formally named clades above families[81], which would leave a limited sample size for statistical analysis.

We recognize that using taxonomy to define clades may impact analyses[95,96] (but also see respective responses[97,98]). To avoid possible biases from clade selection, we also used clades of the same age as alternative units for analysis[96]. We selected age-based clades by considering the most inclusive clade of a given age or younger. With the tree used here[29], a threshold for clade selection much younger than 80 million years would return many groups with few species, limiting variation in net diversification rates. In contrast, a threshold much older than 120 would not return enough clades for robust statistical analysis (e.g., the 120 million-year threshold produced 19 clades; Supplementary Fig. 4). We therefore repeated the radiation-space analyses described below on clades defined by ages of 80, 100, and 120 million years old.

## Quantifying morphological diversity and rates of evolution

We estimated morphological diversity of all anurans, families, age-defined clades, and radiation-space quadrants (see below). We defined morphological diversity as the volume of *n*-dimensional morphospace occupied by a group of species. We used two approaches: a convex-hull volume[35] and a hypervolume[36]. Convex hulls are effectively *n*-dimensional ranges[35]. They likely overestimate shape volume because they are sensitive to outliers and are unable to detect holes—gaps between observations—in *n*-dimensional space[36]. Hypervolume

methods use machine-learning algorithms to determine boundaries around points in *n*-dimensional space and are able to detect and exclude outliers and holes[36,99,100]. Hypervolumes likely underestimate shape and volume depending on the nature of the dataset. For these reasons, the convex hull and hypervolume approaches likely produce a maximal and minimal volume estimate (respectively) of morphological diversity. In consequence, correspondence of results from these two methods should indicate insensitivity to methods of quantifying morphological diversity.

Both methods assume that each axis considered is orthogonal to others, so we used scores from our phylogenetic PCA (pPCA) as data for morphospace calculations. Because both methods are computationally burdensome, we limited analyses to the first five pPC axes. We found in preliminary analyses that five was the best compromise between comprehensiveness and computation time. Moreover, a scree plot (Supplementary Fig. 7) showed a considerable drop in variation explained after five axes[101,102]. These first five axes collectively explained 92.4% of the morphological variation in our dataset (Supplementary Table 1). Most importantly, our results were similar for more (six) and fewer (four) dimensions (Supplementary Table 5). To estimate the convex hull, we used the Quickhull algorithm implemented in the *geometry* package[103], version 0.4.5. To estimate the hypervolume, we used the one-class support vector machine method as implemented in *hypervolume*[100], version 2.0.12.

We estimated multivariate rates of morphological evolution for families and age-defined clades using the method of Adams[37]. This method calculates a single Brownian-motion rate of evolution that accounts for correlations among characters. Brownian motion is the simplest and most general model of continuous-trait macroevolution and is consistent with many different underlying evolutionary scenarios (e.g., stabilizing selection with randomly evolving optima)[46,104,105]. Moreover, previous work has shown that the evolution of these same traits is consistent with a Brownian-motion model in 217 species across many families[1]. Furthermore, given that our sampling of species within families averaged 25% of each family's extant species richness, we emphasize that incomplete clade sampling does not bias this metric. That same previous study[1] (of anurans, with the same traits) used simulations to show that sampling as low as 2.3% of total species diversity has no effect on the accuracy of rate estimation.

We present our raw estimated rates as Supplementary Data 4. However, comparing rates estimated here to previously published rates for other groups is incredibly challenging. While the method we used[37] is increasingly employed for estimating multivariate rates of phenotypic evolution[92], such rate estimates are influenced by different methods for size standardization (e.g., ratios, residuals, General Procrustes Analyses in geometric morphometrics[106]) and different numbers of traits[107].

## Quantifying species diversity and net diversification rates

We followed the classification of AmphibiaWeb[19] for defining families and counting their species diversity. For clades from 80, 100, and 120 million-year time slices, we established species richness using the full tree from Jetz and Pyron[29], which included all known species at the time of their analysis. This tree provides an underestimate of current species richness[19], but this step was necessary to calculate the species diversity of time-sliced clades when genera were separated into multiple clades. It also allowed us to include the species diversity of genera unsampled in the genetic tree of Jetz and Pyron[29], which we used for all other analyses.

We initially estimated net diversification rates using the method-of-moments estimator[38]. This method only requires species richness and clade ages, which are available for all anuran families. Moreover, recent simulation studies show that this method is accurate under many diversification scenarios, including faster rates in younger clades, rate variation over time within clades, and rate heterogeneity

across subclades[68–70]. We recognize that many other methods of calculating diversification rates are available. However, the estimator we used allows as many different rates as families, far more than other methods typically find (e.g., see refs. 108,109). Moreover, this method allows one to estimate the potential effect of extinction on downstream analyses: we can compare how our results (potentially) differ based on low or high extinction fractions. This may be particularly important in anurans, whose oldest families may have low diversity due to high historical extinction rates[110,111]. Yet simple diversification metrics (like the method-of-moments estimator we use) may avoid problems associated with trying to extract too much information from phylogenies of extant taxa[72]. We also emphasize that adaptive radiation may be a temporal phenomenon (i.e., groups characterized as adaptive radiations now may not have been 100 million years ago), as are other macroevolutionary patterns. However, what we see in present-day groups is what we study here: we focus on what led to current species and phenotypic diversity, not how past adaptive radiation led to diversity we no longer see. Thus, using a diversification metric that integrates over the history of clades to the present day is what is most relevant to our study.

We also compared these rates (based on species diversity and ages) with birth-death rates (based on branch lengths) estimated by Moen et al.[1]. Because the birth-death rates could only be estimated for the 38 families with sufficient sampling (at least five species in Jetz and Pyron[29]), we added our originally estimated method-of-moments rates under stem ages and medium extinction fraction for the remaining five families to total 43 families, as in our other diversification-rate analyses. We found that the birth-death rates and method-of-moments rates were highly correlated (Supplementary Table 2). Moreover, our radiation-space results were broadly similar using birth-death rates for diversification (Supplementary Fig. 2). However, we prefer the method-of-moments estimates because we could include all 43 anuran families in this study under a single method of rate estimation.

To be consistent with our morphological analyses, we calculated the stem and crown ages for each family from our MCC consensus tree. Other phylogenies give younger ages for anuran families[81,82]. However, recent diversification analyses using ages from both Jetz and Pyron[29] and Feng et al.[81] showed high correlations in rates across families[1]. For example, rates based on stem ages and an extinction fraction of 0.5 showed a correlation of $r = 0.953$ between the two trees. Here, we calculated rates using three extinction fractions ($\varepsilon$; 0, 0.5, and 0.9), following standard practice[112–114]. We present results based on rates estimated using moderate extinction fractions ($\varepsilon = 0.5$). Low and high extinction fractions gave similar results in downstream analyses (Supplementary Fig. 3). Moreover, we present results based on stem ages, which are estimated from the origination of the clade and are less sensitive to sampling density than crown ages[115]. Results for the latter were highly similar (Supplementary Fig. 3).

## Relationships between rates and diversity

Moen and Wiens[23] showed a strong correlation between species diversity and net diversification rates of anuran families. Here, we re-evaluated this correlation for the 43 families examined in this paper, given updated species richness of families (i.e., >10% of anuran species have been described since 2016; ref. 19). We then tested the relationship between rates of multivariate morphological evolution and morphological diversity across families. We estimated morphological diversity using five-dimensional convex hulls and hypervolumes, as described above. Here, we only examined families with six or more species measured ($n = 27$), because $n + 1$ observations are required to define an $n$-dimensional volume. We then calculated the fifth root of the resulting values. For all variables, we ln-transformed, mean-centered, and scaled them to unit variance (Supplementary Data 5). We then used phylogenetic generalized least-squares (pGLS) correlations to estimate correlations between morphological diversity and

morphological rates of evolution, and net diversification rates and species richness. To be consistent with our calculation of rates, we again used the phylogeny of Jetz and Pyron[29] for our pGLS analyses. However, we expect results to be highly similar with other recent phylogenomic trees[81,82], given that pGLS is highly robust to tree misspecification[116]. We calculated pGLS correlations following Rohlf[117] and using a custom R script from Moen et al.[21].

## Relationships between rates of net diversification and morphological evolution

We next tested the strength of the relationship between rates of net diversification and morphological evolution. This allowed us to examine whether rates were strongly correlated (producing a linear radiation continuum) versus weakly correlated or uncorrelated (yielding a two-dimensional radiation space). We calculated pGLS correlations on the mean-centered and scaled rates of net diversification and morphological evolution ($n = 43$), as described above. We also visualized the relationship between rates by plotting them on the phylogeny with the *ggtree* R package[118], version 3.0.2, with ancestral states estimated by maximum likelihood in *phytools*[91]. Given that we found a weak correlation (see Results), we next describe the continuum along its two dimensions.

## Defining and delimiting the radiation continuum

To characterize an adaptive-radiation space, we separated clades into quadrants by rates of net diversification and morphological evolution, where the origin (0, 0) represented mean values among clades for both rates. Clades with rates of net diversification and morphological evolution >0 were assigned to the adaptive-radiation quadrant. Clades with rates of net diversification and morphological evolution <0 were considered non-adaptive non-radiations. Clades with net diversification rates >0, but rates of morphological evolution <0, were placed in the non-adaptive radiation quadrant. Clades with net diversification rates <0, but rates of morphological evolution >0, were considered adaptive non-radiations.

We also repeated clade assignments after redefining the quadrant boundaries as medians of rates. This alternative scheme allowed us to explicitly examine how robust our results were to quadrant limits. Because all analyses (i.e., families and clades extracted at 80, 100, and 120 million-year time slices) had an odd number of observations, the median clade always straddled at least two quadrants. To avoid omitting any clades, we split clades with median values for either net diversification or morphological rates equally between the quadrants these clades straddled. For morphospace volume calculations (see next section), this meant randomly assigning half (when straddling two quadrants) or a quarter (when straddling all four) of the median-clade species to each quadrant the clade straddled when estimating volumes. For species diversity, we simply divided the number of species in the clade by two (or four) and added them to the number of species observed in the quadrants they straddled.

We then characterized the phylogenetic distribution of evolutionary dynamics (i.e., our four quadrant types) by calculating the $D$ statistic of phylogenetic signal for binary traits[45] as implemented in the *caper* package[119], version 1.0.4. We conducted four analyses, one for each radiation type, with each analysis estimating $D$ for a binary trait consisting of one radiation type versus all others (e.g., for non-adaptive radiation, a trait with one state as "non-adaptive radiation" and the other state as "all other types"). A $D$ of 0 or lower (i.e., negative) would indicate phylogenetic clustering, whereas a $D$ of 1 or higher would indicate a random ($D = 1$) or overdispersed distribution ($D > 1$)[45]. Thus, we tested for a significant deviation from $D = 0.0$ (which would suggest significant random distribution or overdispersion) and from $D = 1.0$ (which would suggest significant clustering). We only conducted this analysis for quadrants delimited by mean evolutionary rates for families, given we found that no quadrant type showed a $D$ significantly

different from 0 or 1 (Supplementary Table 3). We did not expect different results for other ways of characterizing clades or the radiation space.

## Contributions of each radiation type to species diversity and morphological diversity

Our primary goal was to determine the role of adaptive radiation in driving diversity in a major clade. For this goal we needed to first quantify total and quadrant-specific species and morphological diversity, then the proportion of diversity each quadrant of radiation space contained. For species diversity, we tallied the total species richness of the sampled families from AmphibiaWeb[19] to represent total anuran species diversity. This diversity (7359 species) represents >99% of extant described anuran species (7426 species). Thus our results for these 43 anuran families should basically apply to all Anura. We then calculated species diversity for each quadrant by summing the currently described species richness of all families within that quadrant. We divided each quadrant total by the total diversity we analyzed (7359 species) to calculate the proportion of total diversity explained by each of the four types of radiations.

We quantified total morphological diversity as the morphospace volume occupied by all species in our morphological dataset (i.e., the 1226 species for which we could calculate rates of evolution). We then divided the pPC scores into four subsets of species, one subset for each quadrant of the adaptive radiation plane. Each subset included all the species from the clades that we categorized as belonging to that quadrant. We then estimated each quadrant's morphological diversity using the methods described above.

We divided each quadrant's volume by the total anuran volume to calculate the contribution of each radiation type to total anuran morphological diversity. Unlike species diversity, where each quadrant's species contribute independently to total species diversity, morphospaces of different quadrants may overlap. When this occurs, the sum of quadrant percentages may total more than 100%. Alternatively, quadrant percentages may not sum to 100% if quadrant morphospaces occur in mutually exclusive regions of the total anuran morphospace (i.e., gaps between quadrants within the total anuran morphospace)[99].

## Time-independent rates of net diversification and morphological evolution

Both net diversification rates and rates of phenotypic evolution include time in their estimation. While time is directly used in the calculation of net diversification rate, it is involved in morphological rates through phylogenetic branch lengths. Such a shared dimension could, in principle, lead to similarity in these two types of rates (e.g., a family with a high net diversification rate could have a high rate of phenotypic evolution). Moreover, both rates often show a negative relationship with time across many groups of organisms[120]. Thus, we further explored the potential effect of shared time on our net diversification and multivariate morphological rate estimates. For brevity, we circumscribed these analyses to include only net diversification rates estimated using stem ages and moderate extinction fractions ($\varepsilon = 0.5$). First, we assessed the relationship between age and rate by using phylogenetic generalized least squares (pGLS) regression under Brownian motion, as implemented in the R package *phylolm*, version 2.6.2[121]. We regressed net diversification rates on stem age (i.e., rather than crown age) because it was the age used to calculate the rates on which we focused here. In contrast, we regressed rates of phenotypic evolution on crown age, given that only the crown phylogeny of each family was used for estimating rates of evolution (using stem ages led to even weaker relationships). These regressions showed weak but statistically significant relationships between each rate and their respective family age estimates. Surprisingly, morphological rate of evolution had a significant positive slope ($\beta = 0.014 \pm 0.006$;

$R^2_{Adj} = 0.077$; $P = 0.040$), contrasting with the typically negative relationship[122–124]. Net diversification rate showed a significant negative relationship with time ($\beta = -0.020 \pm 0.005$; $R^2_{Adj} = 0.231$; $P < 0.001$), as expected when regressing a ratio against its denominator[125,126].

We then assessed whether time-independent net diversification rates and morphological rates of evolution were correlated. We did this by calculating residuals from each of the regression models; such residuals represent time-independent measures of net diversification rate and morphological rate of evolution. We examined the correlation with pGLS, as in our other correlation analyses. Similar to our main correlation analyses, which did not account for time explicitly, time-independent rates were uncorrelated ($r = 0.035$; $P = 0.825$).

### Reporting summary
Further information on research design is available in the Nature Portfolio Reporting Summary linked to this article.

## Data availability
All data, including raw intraspecific morphological data, species means, microhabitat states, and phylogenies are available as Supplementary Data. They have also been permanently archived and are available on the Dryad Digital Repository (https://doi.org/10.5061/dryad.hx3ffbggp)[127]. Data on microhabitats (Supplementary Data 1) were in part gathered from the publicly available web databases AmphibiaChina (http://www.amphibiachina.org/), AmphibiaWeb (http://amphibiaweb.org), Anfibios del Ecuador (https://bioweb.bio/faunaweb/amphibiaweb/), and IUCN (http://www.iucnredlist.org).

## Code availability
All custom analysis code has been included as Supplementary Code 1. It has also been permanently archived and is available on Zenodo (https://doi.org/10.5281/zenodo.8422404).

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

## Acknowledgements

We thank K. Adams, H. Brew, M. Caron, H. Dupire, A. Fery, A. R. Hanna, L. Lacy, E. Mendoza, B. Rae, E. Shore, S. Starr, M. Stevens, A. Vargas, C. Slattery, J. Spicer, A. Van Pelt, S. P. Vijayakumar, M. Wisdom, and A. Zakrzewicz for help with data collection. This project would not have been possible if not for the logistical support, patience, and generosity of the curators, collections managers, and staff of natural history collections across the U.S. We thank C. Spencer, M. Koo, and J. McGuire (Museum of Vertebrate Zoology, Univ. California Berkeley); K. de Queiroz, R. Wilson, A. Wynn, C. Keating Sami, and K. Tighe (Smithsonian National Museum of Natural History); B. Stuart and J. Beane (North Carolina State Museum of Natural Sciences); A. Resetar, R. Grill, and J. Mata (Field Museum); R. Brown, R. Glor, and L. Welton (University of Kansas Biodiversity Institute); G. Pauly and N. Camacho (Los Angeles County Natural History Museum); J. Vindum, L. Scheinberg, and E. Ely (California Academy of Sciences); F. Burbrink, C. Raxworthy, D. Kizirian, M. Arnold, and L. Vonnahme (American Museum of Natural History); G. Schneider and D. Rabosky (University of Michigan Museum of Zoology); J. Rosado and J. Losos (Museum of Comparative Zoology, Harvard Univ.); C. Franklin and J. Campbell (Amphibian and Reptile Diversity Research Center, Univ. Texas Arlington); K. McBee, D. Lynch, and J. Agan (Oklahoma State University Collection of Vertebrates); M. Hagemann (Bernice Pauahi Bishop Museum); C. Austin and S. Parker (Louisiana State University Museum of Natural Science); C. Siler and J. Watters (Sam Noble Museum, Univ. Oklahoma); T. LaDuc (University of Texas at Austin Biodiversity Center); D. Blackburn and C. Sheehy (Florida Museum of Natural History); and S. Rogers and S. Kennedy-Gold (Carnegie Museum of Natural History). We thank Oklahoma State University and the National Science Foundation (awards DEB-1655812 to D.S.M. and DEB-1655690 to J.J.W.) for financial support.

## Author contributions

G.M., J.J.W., and D.S.M. conceived of the study. G.M. and D.S.M. collected the data. G.M. analyzed the data and produced the figures. G.M., J.J.W., and D.S.M. wrote the paper. The manuscript reflects the contributions and ideas of all authors.

## Competing interests
The authors declare no competing interests.

## Additional information

**Peer review information** : *Nature Communications* thanks Margot Michaud and the other, anonymous, reviewer(s) for their contribution to the peer review of this work. A peer review file is available.

