## [Peer Review File · Nature Communications]

The radiation continuum and the evolution of frog diversityReviewers' Comments:

Reviewer #1:

Remarks to the Author:

The manuscript investigates the macroevolutionary dynamics within *Aura* species with a special focus on adaptive radiation-like patterns. Using a new well-explained and straightforward work frame the authors identified a variety of macroevolutionary combinations of low and high rates of phenotypic evolution and diversification. They highlight the importance of adaptive radiation events explaining the morphological and taxonomical diversity in frogs while emphasizing the relevance of describing other evolutionary patterns (e.g. non-adaptive radiations, adaptive non-radiations) to understand our current biodiversity.

The sample size is impressive, the manuscript is well-written, and the authors seem aware of the limitations of their study and the concept of adaptive radiation. I think that the manuscript has all the quality criteria to match the level of Nature portfolio's expectations. I only have a few comments and suggestions which would improve the overall quality of the manuscript and figures. I propose -but not required - to add some analyzes to better describe macroevolutionary patterns observed in this study and improve some figures. In addition, some important methodological points need to be clarified.

Below I provide line-by-line comments:

Abstract

L.9-10. Yet its contribution to species and phenotypic diversity of a major group has not been examined.

It would be interesting to specify here what this group is, and why it is "major".

Introduction

L.24-25. Adaptive radiation is characterized by high rates of species diversification and phenotypic evolution.

True, but it is important to add a fundamental nuance here: adaptive radiations are characterized by high rates of species diversification and phenotypic evolution compared to close relative taxa (see Losos and Miles, 2002). This may seem trivial, but it is not. As you mentioned in lines L.138-141, It is important to remember that the evolutionary phenomenon of adaptive radiation can only be described within a specific phylogenetic framework.

L.40-42. Yet, to our knowledge, the phenotypic and species diversity represented by these different types of clades has not been quantified [...].

This sentence is a bit confusing. Are you saying that we have never yet assessed the importance of these evolutionary phenomena (non-adaptive radiations and adaptive non-radiations) on life's diversity?

L.51-52. Importantly, diversification rates are correlated with species richness across families.

Can you further emphasize why this is important in the context of your study?

L.74. [...] including 51 of 54 families.

It seems that you use 2 datasets for methodological reasons: one based on 51 families and one of 43 families, yet it is not clear which dataset is used for which analyses. You really should clarify this through the manuscript, and explain why not just do all the analyzes on n=43 families in order to have consistent results on the same dataset

Results

Fig. 1. Please add in the description that the blue dot represents the position of your species in the morphospace and that the grey hexagons represent the total density of the species within Anura order in the morphospace.

Fig. 2. Please add the PGLS regression slopes regression for the HV and CH and if possible the confidence interval

L.115. Diversity and the radiation continuum

I think that you should add in figure 4 what a radiation continuum should look like under this conceptual framework (see what has been done by Felice et al. 2018).

L.131-134. However, limiting the term "adaptive radiation" only to clades that show significantly elevated rates would mean that most clades have unexceptional rates of diversification and morphological evolution.

Beware of this reasoning. One could easily believe that you decided to base your analyses and conclusions on the quadrants rather than other adaptive radiation definitions only because the latter did not fit with your initial assumptions and does not give you the expected results (which is a major scientific conceptual bias in setting up a study).

L.177-189. This could happen if a clade with a low rate of morphological evolution generates more morphological diversity simply by having more species or being older.

Does this assumption match what you observe for your data? If so, can you describe this pattern and the families concerned?

L.180-190. Second, some clades with low rates of morphological evolution may represent distinct parts of anuran morphospace [...] For example, the non-adaptive radiation quadrant includes clades that are mostly aquatic (Ranidae, Telmatobiidae), arboreal (Centrolenidae, Hemiphractidae, and Rhacophoridae), and terrestrial (Leptodactylidae). These ecomorphs collectively span anuran morphospace [...]

Have you considered using PGLS analyses to test whether the morphotypes associated with these broad ecological categories represent distinct places in the morphospace?

L.203-207. Calculating net diversification rates with birth-death estimators produced very similar results (Supplementary Fig. 2). Using crown ages and different extinction fractions also resulted in similar proportions of species diversity and morphological diversity among quadrants (Supplementary Fig. 3).

Would be nice to have a quick description of the main differences here.

Methods

L.352. We measured 4,628 adult specimens of 1,234 species from around the world.

Could you clarify if these measurements were made on living animals (wild specimens? captive specimens?) or only on museum collections?

L.358-360. However, for eight families we were only able to sample a single species, which prevented calculating rates of morphological evolution.

As recommended by Adams et al. (2009), you should only consider taxonomic groups with four or more species to avoid problems of morphological evolution rate estimation: "First, to avoid problems in estimating rates of change among very few species, only clades with >4 sampled species were used; clades with fewer species were subsumed into larger clades (given the requirement of monophyly) or otherwise not used".

L.361-363. To reduce potential effects of sexual-size dimorphism on our sampling 75-77, we measured male specimens when possible (89% of all specimens sampled).

It would be more relevant to provide the percentage of species where you measured both males and females.

L.365-372. We quantified body shape using linear, area, and volumetric measurements of traits that are ecologically and functionally relevant to locomotion and microhabitat use. First, we measured snout-vent length, head length, head width, upper arm, forearm, hand, thigh, crus, tarsus, and foot lengths to the nearest 0.01 mm using a Mitutoyo digital caliper (Kanagawa, Japan). Second, we photographed the foot and hand of each specimen and measured the areas of digit tips on both the front and hind limb, interdigital webbing of the hind limb, and the inner metatarsal tubercle using ImageJ.

Were the measurements taken several times and then averaged? Also, it would be nice to have a figure to show these measurements on a specimen or a schematic illustration.

L.421-422. We summarized diversity in body form using a phylogenetic principal components analysis (PCA) on species means

Shouldn't this be a pPCA? Also, it's not clear what metrics you used to make this pPCA. In the introduction, you talk about 10 ecologically relevant morphological traits, but you describe 18 phenotypic variables in the section and SI-table 3.

L.426-428. Using a Mantel test (10,000 permutations), as implemented in the package *vegan* version 2.5.7, we tested to ensure that eigenvectors from phylogenetic and non-phylogenetic PCA were strongly correlated ($r_{\text{Mantel}}=0.885$; $P<0.001$).

This sentence is not complete.

L.441-442. We recognize that using taxonomy to define clades may impact analyses.

There are now methods to detect shifts in evolutionary rates without a priori assumptions on the taxonomic level (see *RRPhylo* package; Castiglione et al. 2018). Such methods are easy to implement with your dataset and could greatly support your results.

L.471-472. Moreover, a scree plot showed a considerable drop in variation explained after five axes. Please, provide this graph in the SI.

L.479-480. We estimated multivariate morphological rates of evolution for families and age defined clades using the methods of Adams.

As said above, if you use the method proposed by Adams, it is crucial to consider only taxonomic groups with at least 4 species. It would be good to inform here how many families / taxonomic groups could be analyzed for the estimation of phenotypic rates of evolution within your dataset.

L.580. Defining and delimiting the radiation continuum

Figures 3 and 4 are good, but the manuscript would gain in clarity and would be more impactful if you added a figure showing the different results of the evolutionary quadrats that you were able to define showing which groups fit into different evolutionary modalities and highlighting events of adaptive radiation directly on your phylogeny.

Supplementary Information

Supplementary Fig. 2. Please add that gray dots are families whose volume could not be estimated.

L.980. Muscle mass and volume

It is never explained what this variable is used for. Is it used to define the ecomorphology of species? Is it part of the 10 measures ecologically relevant morphological traits mentioned in the introduction?

References:

Adams, D. C., Berns, C. M., Kozak, K. H. & Wiens, J. J. Are rates of species diversification correlated with rates of morphological evolution? *Proc. R. Soc. B Biol. Sci.* (2009) doi:10.1098/rspb.2009.0543.

Castiglione, S. et al. A new method for testing evolutionary rate variation and shifts in phenotypic evolution. *Methods Ecol. Evol.* 9, 974–983 (2018).

Felice, R. N., Randau, M. & Goswami, A. A fly in a tube: Macroevolutionary expectations for integrated phenotypes. *Evolution (N. Y.)*. (2018) doi:10.1111/evo.13608.

Losos, J. B. & Miles, D. B. Testing the hypothesis that a clade has adaptively radiated: Iguanid lizard clades as a case study. *Am. Nat.* 160, 147–157 (2002).

Reviewer #2:

Remarks to the Author:

Review of 'radiation continuum'

This is an interesting paper that should find an audience among evolutionary biologists interested in adaptive radiation. The overarching result is unsurprising ("all four major types of macroevolutionary outcomes are represented"), and some potentially major findings ("Less than half of frog families resembled adaptive radiations") are contingent on the subjective identification of both units for analysis and other elements, including the definition of adaptive radiation itself. The authors are not unaware of these subjective factors—these are problems that characterize many comparative studies and most studies of diversification and adaptive radiation—and the authors take steps to attempt to address many of them, performing multiple analyses with parameters tweaked. But their conclusions are sensitive to these perturbations, and there is only so much potential variation that can be accounted for before the number of necessarily arbitrary decisions renders even broadstroke inferences shaky. That all being said, the characterization of a "radiation space" is a strong contribution.

Here are some of the subjective/problematic issues of this paper: taxonomic families are not suitable units for analyses; use of morphological characterization of clades from a tiny percentage of sampled species is imprudent for 'phenotypic diversification' analyses; speciation/diversification analysis of clades from which a small percentage of species is included is even less prudent than phenotypic analyses that undersample phenotypic diversity; adaptive and nonadaptive radiations/nonradiations occur along a continuum so any classification of a lineage or group of lineages as an adaptive radiation (AR) necessarily is the result of arbitrary cutoffs regarding these terms.

All of the above issues have stock solutions in print (e.g., for undersampling of species for species diversification analyses: simulation studies, where undersampling of a known distribution results in a reasonable estimate of that known distribution [surprise!], is interpreted as informative regarding reality)—and, again, the authors are not incognizant of these problems. But unfortunately some of these issues are, in my view, insoluble (subjective cutoffs to reach AR) and others simply must await more thorough sampling before they can be applied informatively on this scale (i.e., the scale of all frogs).

More on some of these points below:

--Families are not appropriate units of comparative evolutionary analysis, for two reasons: 1) they are arbitrary units, 2) they are not independent.

The authors are aware of the arbitrariness issue and attempt steps to counter this concern by comparing clades of equal age. However, implementation of this admirable safeguard seems to have a major effect on conclusions. The use of families (in my view, the least desirable of clade samples that were used—why would we consider human designations in analyses of the natural world?) returns the '75% of diversity found in 50% of AR clades' result that is reported in the abstract as a (the?) main

result—numbers that are used to argue that adaptive radiations "... clearly do not explain nearly all diversity, as often posited." But use of a 120 mya cutoff for clades results in 96/98% diversity occurring in "adaptive-radiation like clades," which would certainly seem consistent with the idea that adaptive radiations explain "nearly all diversity." Which of these, or other possibilities, is the "correct" answer depends on an arbitrary decision regarding the units of analysis.

Regarding point 2), I may have missed it, but it does not appear to me that any phylogenetic "correction" was implemented for the interrelationships of the families. I realize it is not standard practice to account for phylogenetic covariance in macroevolutionary comparisons of clades; but it should be—diversification rates, like anything else, may display some phylogenetic inertia that must be accounted for (if not, it undermines the analysis of clades at all). Rhacophoridae and Mantellidae share some 80 million years of evolution separate from Heleophrynidae. Wouldn't this shared history suggest some propensity for shared diversification rates in these two families? If you are going to compare across a group of clades (which itself is problematic), minimally you must account for the nonindependence of clades being compared.

--The major conclusions of this paper are contingent on the structure (statistically, the distribution) of the "radiation continuum" (lines 581-585). The construction of this space is interesting and should be useful for visualizing adaptive radiations. But the details of how this space may be useful for hypothesis testing, or even statements of relative frequency, remain to be worked out. Perhaps the construction/discussion of this space is interesting enough for this paper to be accepted—making AR quantitative as a comparative concept is a big deal. But, the framing of this paper as some "test" of adaptive radiations, when this test rests on arbitrary cutoffs of what constitutes an AR, is not compelling.

Perhaps it would help to create a null "radiation space" using simulations where species diversification and morphological evolution are simultaneously modeled using simple (Brownian-motion like) models. This paper's current characterization of this space as quadrants centered around means from empirical data is great for visualization but not so great for assessing the relative frequency of patterns in nature.

lines 10-13

"what proportion of clades show macroevolutionary dynamics similar to adaptive radiations? Second, what proportion of overall species richness and phenotypic diversity do these adaptive radiation-like clades contain?"

These questions are inescapably connected to arbitrary decisions regarding what constitutes an "adaptive radiation" (AR)—that is, how rapidly must speciation occur before it is rapid enough to potentially qualify as an AR. For example: couldn't the results of this paper be spun as new evidence that our definition of AR needs revision, rather than as evidence that ARs do or don't explain diversity as expected? There are no hypotheses to test here; just patterns to report (not that that disqualifies the work scientifically).

Lines 132-134

" However, limiting the term "adaptive radiation" only to clades that show significantly elevated rates would mean that most clades have unexceptional rates of diversification and morphological evolution."

This statement epitomizes the issues with subjectivity/perception for these results. What if the real evolutionary pattern in nature is that "most clades have unexceptional rates of diversification and morphological evolution." The characterization of adaptive radiation in this paper apparently disallows this result.

lines 953-55

Aren't the 11 omitted families critical datapoints? I.e., cases where near-maximally low diversification rates occur? (also, it appears that Ascaphidae, which includes only two species, was included in

analysis. Were the 11 excluded families monotypic?).

Some summary comments:

Methods

The methods used are standard for the field, i.e., they have been utilized in studies published in prestigious journals. In my view, there are insurmountable issues for some of these approaches—in particular, the practices of using families as units of evolutionary analysis and of measuring diversification rate in severely species-undersampled clades should both be retired.

Importance

The characterization and visualization of a "radiation space" seems a major contribution. The conclusions regarding the frequency of "adaptive radiation" are not compelling.

Writing

The writing is mostly clear and easy to follow. There are some general statements that come off as straw-man ("Most of life's vast diversity of species and phenotypes is often attributed to adaptive radiation").

Figures

Good, effective.

Reviewer #3:

Remarks to the Author:

This manuscript quantifies morphological diversity, morphological rates of evolution, species diversity, and net diversification rates across frog families to define and delimit a radiation space. This novel radiation continuum method is then used 1) to identify the proportion of anuran clades exhibiting macroevolutionary dynamics indicative of adaptive radiations with high rates of both morphological evolution and net diversification and 2) to quantify the proportion of total frog species richness and phenotypic diversity contained within these adaptive radiation-like clades. The authors demonstrate that less than half of frog families fall within the "adaptive-radiation-like" quadrant, yet these lineages represent most diversity. The results presented here will be of broad interest to biologists, and I think this framework will have an important impact on the study of adaptive radiations and quantifying macroevolutionary patterns. I commend the authors for their robust statistical approaches, which included multiple methods and parameter sensitivity analyses for quantifying morphology, diversification rates, and their radiation space. All code and data are available and well organized. I have a few questions and comments for the authors.

My only concern related to analyses is whether the method used to calculate morphological rates (compare.evol.rates) is sensitive to variability in number of species across groups? I generally find the patterns identified in the radiation space for sampled families intuitive as a frog biologist, but I am surprised by some of the outlier families with high rates of morphological evolution. Most of these outlier families have low species richness and are therefore represented by only a few species in the dataset (e.g., Cycloramphidae, 5 species; Pelodytidae, 2 species; Petropedetidae, 5 species; Alytidae, 5 species). It seems unlikely these families have higher morphological rates of evolution than Bufonidae or Microhylidae, for example. The families with the lowest rates of morphological evolution are also represented by only a few species, but this pattern is less surprising.

Would it be possible to compare the family-level radiation space to the clade age radiation space by mapping the adaptive radiation, non-adaptive radiation, adaptive non-radiation, and non-adaptive

non-radiation quadrants for each approach as traits to the tips of your phylogeny? I recognize that the adaptive-radiation quadrant still included most species diversity and morphological diversity using the clade age approach, but I am wondering how the species composition of the four quadrants have shifted (i.e., are the species within Ceratophryidae still within the NANR quadrant for the 80 my clade age analysis?).

I think it would generally be interesting to visualize the phylogenetic distribution of the radiation space quadrants to assess, for example, whether adaptive radiation-like families are clustered or dispersed across the frog tree of life.

Reviewer #1 (Remarks to the Author):

The manuscript investigates the macroevolutionary dynamics within *Aura* species with a special focus on adaptive radiation-like patterns. Using a new well-explained and straightforward work frame the authors identified a variety of macroevolutionary combinations of low and high rates of phenotypic evolution and diversification. They highlight the importance of adaptive radiation events explaining the morphological and taxonomical diversity in frogs while emphasizing the relevance of describing other evolutionary patterns (e.g. non-adaptive radiations, adaptive non-radiations) to understand our current biodiversity.

The sample size is impressive, the manuscript is well-written, and the authors seem aware of the limitations of their study and the concept of adaptive radiation. I think that the manuscript has all the quality criteria to match the level of Nature portfolio's expectations. I only have a few comments and suggestions which would improve the overall quality of the manuscript and figures. I propose -but not required - to add some analyzes to better describe macroevolutionary patterns observed in this study and improve some figures. In addition, some important methodological points need to be clarified.

Response: We thank the reviewer for their positive feedback. We think that the manuscript has been improved thanks to these comments.

Below I provide line-by-line comments:

Abstract

L.9-10. Yet its contribution to species and phenotypic diversity of a major group has not been examined.

It would be interesting to specify here what this group is, and why it is "major".

Response: We have changed this phrase to include "...major group (such as an old clade with thousands of species)". We understand that there can be some ambiguity in what constitutes a "major" group, and that makes defining the term challenging, so we give this example of some key characteristics. We hesitate to more thoroughly justify the term in the abstract, where space is limited. However, as we analyze 1,226 species (which we state just two sentences later) and Anura has >7,500 species (which we state in the introduction), we think that most readers will agree that this is a "major" group.

Introduction

L.24-25. Adaptive radiation is characterized by high rates of species diversification and phenotypic evolution.

True, but it is important to add a fundamental nuance here: adaptive radiations are characterized by high rates of species diversification and phenotypic evolution compared to close relative taxa (see Losos and Miles, 2002). This may seem trivial, but it is not. As you

mentioned in lines L.138-141, It is important to remember that the evolutionary phenomenon of adaptive radiation can only be described within a specific phylogenetic framework.

Response: We agree with the reviewer—any consideration of adaptive radiation must be through a phylogenetic comparative framework. This is the impetus for our study. Throughout the manuscript, particularly early on (refs. 1,6–15), we reference works that either show or argue this point.

L.40-42. Yet, to our knowledge, the phenotypic and species diversity represented by these different types of clades has not been quantified [...].

This sentence is a bit confusing. Are you saying that we have never yet assessed the importance of these evolutionary phenomena (non-adaptive radiations and adaptive non-radiations) on life's diversity?

Response: Yes—as far as we know, no work has quantified phenotypic or species diversity produced by these different evolutionary dynamics (e.g., non-adaptive radiation, adaptive non-radiation). To clarify this point, we have changed the wording here from “...represented by these different types of clades...” to “produced by these different evolutionary dynamics...”.

L.51-52. Importantly, diversification rates are correlated with species richness across families. Can you further emphasize why this is important in the context of your study?

Response: We have added, “However, it is unclear whether species-rich clades with high diversification rates are also morphologically diverse” after the sentence noted by the reviewer. We added this statement to clarify both what is known from previous studies and also what the existing knowledge gap is.

L.74. [...] including 51 of 54 families.

It seems that you use 2 datasets for methodological reasons: one based on 51 families and one of 43 families, yet it is not clear which dataset is used for which analyses. You really should clarify this through the manuscript, and explain why not just do all the analyzes on $n=43$ families in order to have consistent results on the same dataset

Response: We have now clarified the language where relevant and we now include sample sizes wherever they were not present in the previous submission. To economize the main text, we note that most such justification is in the methods (i.e., later in the manuscript), including new text we have added in response to this comment. For example:

“...with mean=0.00131 ($n=43$; Dataset S1)”

“...scaled rates of net diversification and morphological evolution ($n=43$)...”

Results

Fig. 1. Please add in the description that the blue dot represents the position of your species in the morphospace and that the grey hexagons represent the total density of the species within Anura order in the morphospace.

Response: Done.

Fig. 2. Please add the PGLS regression slopes regression for the HV and CH and if possible the confidence interval

Response: The analyses shown in Figure 2 are correlations, not regressions. Thus, no slopes are estimated. Here, the correlations that are calculated take into account covariance among tips in the same way that pGLS regression does, hence the name of pGLS correlation.

L.115. Diversity and the radiation continuum

I think that you should add in figure 4 what a radiation continuum should look like under this conceptual framework (see what has been done by Felice et al. 2018).

Response: We greatly appreciate the reviewer's point here. Therefore, we tried different potential layouts that would include these hypothetical elements in addition to the observed data. Unfortunately, our conclusion from these attempts was that adding such hypothetical elements would make the figure quite confusing. Furthermore, while it is simple to show a highly correlated radiation spectrum (i.e., along a single dimension), it is unclear what exactly a hypothetical distribution of the uncorrelated rates should look like. Specifically, there are many different distributions of these rates that could potentially show weakly correlated or uncorrelated axes.

L.131-134. However, limiting the term "adaptive radiation" only to clades that show significantly elevated rates would mean that most clades have unexceptional rates of diversification and morphological evolution.

Beware of this reasoning. One could easily believe that you decided to base your analyses and conclusions on the quadrants rather than other adaptive radiation definitions only because the latter did not fit with your initial assumptions and does not give you the expected results (which is a major scientific conceptual bias in setting up a study).

Response: We strongly agree that, out of context, this sentence raises concerns about "moving the goal posts" to fit a narrative. However, we qualify the sentence by stating reasons why we define the quadrants to be more like one type than another in the sentences that follow it. Namely, we state that such quadrant boundaries can be defined in various ways. Moreover, quadrant boundaries are likely to differ depending on the group of organisms that are examined.

L.177-189. This could happen if a clade with a low rate of morphological evolution generates more morphological diversity simply by having more species or being older. Does this assumption match what you observe for your data? If so, can you describe this pattern and the families concerned?

Response: We now point out the examples of Batrachylidae, Phrynobatrachidae, and Ptychadenidae, all of which could potentially represent this phenomenon.

L.180-190. Second, some clades with low rates of morphological evolution may represent distinct parts of anuran morphospace [...] For example, the non-adaptive radiation quadrant includes clades that are mostly aquatic (Ranidae, Telmatobiidae), arboreal (Centrolenidae, Hemiphractidae, and Rhacophoridae), and terrestrial (Leptodactylidae). These ecomorphs collectively span anuran morphospace [...]

Have you considered using PGLS analyses to test whether the morphotypes associated with these broad ecological categories represent distinct places in the morphospace?

Response: We greatly appreciate this suggestion. However, several recent studies have focused specifically on testing whether these microhabitat specialists are morphologically distinct (e.g., Moen et al., 2013, 2016, 2021), including studies that also incorporated performance data (Moen et al., 2013, 2021). We cite these papers already, and we think it would be redundant to repeat these analyses here.

L.203-207. Calculating net diversification rates with birth-death estimators produced very similar results (Supplementary Fig. 2). Using crown ages and different extinction fractions also resulted in similar proportions of species diversity and morphological diversity among quadrants (Supplementary Fig. 3).

Would be nice to have a quick description of the main differences here.

Response: We now provide a brief description of the differences between the analyses we present in the main manuscript and those found in the supplement.

Methods

L.352. We measured 4,628 adult specimens of 1,234 species from around the world. Could you clarify if these measurements were made on living animals (wild specimens? captive specimens?) or only on museum collections?

Response: Thank you for pointing out this confusion. We have now rephrased to say “4,628 adult, museum specimens of...” to clarify that we only measured museum (i.e., preserved) specimens.

L.358-360. However, for eight families we were only able to sample a single species, which prevented calculating rates of morphological evolution.

As recommended by Adams et al. (2009), you should only consider taxonomic groups with four or more species to avoid problems of morphological evolution rate estimation: “First, to avoid problems in estimating rates of change among very few species, only clades with >4 sampled species were used; clades with fewer species were subsumed into larger clades (given the requirement of monophyly) or otherwise not used “.

Response: In response to this important comment, we have now mentioned how the results would be impacted by excluding the 11 clades with 2–4 species sampled, in the section of the Methods titled “Morphological data collection.” Nevertheless, we think that it is important to note that the study by Adams et al. (2009) did not give any justification or rationale for the cut-off of four species. Furthermore, they provided no evidence that rate estimation would be problematic for clades with this number of species. In our dataset, there are 11 families that are included based on sampling of 2 to 4 species, and some of these families do not have more than 4 extant species (i.e., even with complete sampling). Thus, we think it is essential to include these clades, because excluding them would bias our study towards characterizing only more species-rich clades, whereas we wanted to characterize the full spectrum of species diversity across anuran families.

L.361-363. To reduce potential effects of sexual-size dimorphism on our sampling⁷⁵⁻⁷⁷, we measured male specimens when possible (89% of all specimens sampled).

It would be more relevant to provide the percentage of species where you measured both males and females.

Response: We now mention in the parenthetical statement that 82% of sampled species were represented solely by males.

L.365-372. We quantified body shape using linear, area, and volumetric measurements of traits that are ecologically and functionally relevant to locomotion and microhabitat use. First, we measured snout-vent length, head length, head width, upper arm, forearm, hand, thigh, crus, tarsus, and foot lengths to the nearest 0.01 mm using a Mitutoyo digital caliper (Kanagawa, Japan). Second, we photographed the foot and hand of each specimen and measured the areas of digit tips on both the front and hind limb, interdigital webbing of the hind limb, and the inner metatarsal tubercle using ImageJ.

Were the measurements taken several times and then averaged? Also, it would be nice to have a figure to show these measurements on a specimen or a schematic illustration.

Response: We only took each measurement once, as our measurements were highly precise—our preliminary repeated measurements showed a coefficient of variation of less than 0.03 (i.e., the standard deviation was less than 3% of the mean) for all measurements, with most <0.015.

In terms of the measurements themselves, we agree that a figure could be nice, but we worry it would suffer from low visual precision (e.g., it can be hard to see the precise anatomical location of some measurement on a two-dimensional drawing, whereas our verbal description indicates precisely where to place the calipers). Thus, we instead provide a detailed description of the measurements in Supplementary Table 4.

L.421-422. We summarized diversity in body form using a phylogenetic principal components analysis (PCA) on species means

Shouldn't this be a pPCA? Also, it's not clear what metrics you used to make this pPCA. In the introduction, you talk about 10 ecologically relevant morphological traits, but you describe 18 phenotypic variables in the section and SI-table 3.

Response: Yes, “pPCA” and “phylogenetic PCA” are the same thing, just with a slightly different acronym. We prefer to retain "PCA" as our acronym and specify (where relevant) whether a given analysis was phylogenetic or not, because we discuss both versions of this procedure in this paragraph.

Among the 18 traits we describe, seven of them are summed for the front and hindlimb lengths (three of the fore limb, four of the hindlimb). SVL is used for size-correction, but is not in the phylogenetic PCA, leaving 10 traits for the latter. We now state in the Methods section that the respective limb elements were summed together. We also now explicitly state what measurements were used in the phylogenetic PCA.

L.426-428. Using a Mantel test (10,000 permutations), as implemented in the package *vegan* version 2.5.7, we tested to ensure that eigenvectors from phylogenetic and non-phylogenetic PCA were strongly correlated ($r_{\text{Mantel}}=0.885$; $P<0.001$).

This sentence is not complete.

Response: We have clarified this sentence. It now reads “We compared the results of these two types of PCA by conducting a Mantel test (10,000 permutations) on the PCA eigenvectors, as implemented in the package *vegan*⁹¹ version 2.5.7. This analysis showed a strong correlation ($r_{\text{Mantel}}=0.885$; $P<0.001$) between phylogenetic and non-phylogenetic PCAs.”

L.441-442. We recognize that using taxonomy to define clades may impact analyses.

There are now methods to detect shifts in evolutionary rates without a priori assumptions on the taxonomic level (see RRPhylo package; Castiglione et al. 2018). Such methods are easy to implement with your dataset and could greatly support your results.

Response: We thank the reviewer for this suggestion. It is an intriguing idea, but the analysis would not really be practical for our study, for which we need to define clades *a priori* and compare their rates of diversification and phenotypic evolution. Defining clades based solely on shifts in their phenotypic rates would be problematic. We think that the clades need to be defined in a way that does not favor rates in one type of

variable over another. We also note that alternative ways of defining clades (i.e., based on clade age) produced largely similar results.

L.471-472. Moreover, a scree plot showed a considerable drop in variation explained after five axes.

Please, provide this graph in the SI.

Response: We have now provided the scree plot as Supplementary Figure 7. We also note that we included it in the data-analysis tutorial, which was part of the supplementary materials of the paper.

L.479-480. We estimated multivariate morphological rates of evolution for families and age defined clades using the methods of Adams.

As said above, if you use the method proposed by Adams, it is crucial to consider only taxonomic groups with at least 4 species. It would be good to inform here how many families / taxonomic groups could be analyzed for the estimation of phenotypic rates of evolution within your dataset.

Response: We addressed this point above in response to a previous, similar comment. Note that the paper by Adams et al. (2009) did not use the multivariate rate methods utilized here, nor did Adams (2014) specify any lower limits of sampling to use his method. However, Moen et al. (2021) used simulations to show that the method of Adams (2014) is unbiased by sampling only a small fraction (2.3%) of the species in a clade. We now point this out in the Methods, in the section titled “Quantifying morphological diversity and rates of evolution”.

L.580. Defining and delimiting the radiation continuum

Figures 3 and 4 are good, but the manuscript would gain in clarity and would be more impactful if you added a figure showing the different results of the evolutionary quadrats that you were able to define showing which groups fit into different evolutionary modalities and highlighting events of adaptive radiation directly on your phylogeny.

Response: In response to this suggestion, we have now added in Fig. 3 the types of radiation each family represents as tip labels. This also accommodates a similar comment by Reviewer 3, who wanted to see the phylogenetic distribution of families in the radiation space. We hope this change satisfies Reviewer 1's desire to see where on the phylogeny the adaptive-radiation-like clades occur.

Supplementary Information

Supplementary Fig. 2. Please add that gray dots are families whose volume could not be estimated.

Response: Done.

L.980. Muscle mass and volume

It is never explained what this variable is used for. Is it used to define the ecomorphology of species? Is it part of the 10 measures ecologically relevant morphological traits mentioned in the introduction?

Response: Yes, muscle mass is one of the phenotypic variables for describing the ecomorphology of species. In response to this comment, we now state this explicitly at the beginning of this supplementary methods sections. We gave more details about this particular variable in the methods (lines 384–385, 396–399).

References:

Adams, D. C., Berns, C. M., Kozak, K. H. & Wiens, J. J. Are rates of species diversification correlated with rates of morphological evolution? *Proc. R. Soc. B Biol. Sci.* (2009) doi:10.1098/rspb.2009.0543.

Castiglione, S. et al. A new method for testing evolutionary rate variation and shifts in phenotypic evolution. *Methods Ecol. Evol.* 9, 974–983 (2018).

Felice, R. N., Randau, M. & Goswami, A. A fly in a tube: Macroevolutionary expectations for integrated phenotypes. *Evolution* (N. Y). (2018) doi:10.1111/evo.13608.

Losos, J. B. & Miles, D. B. Testing the hypothesis that a clade has adaptively radiated: Iguanid lizard clades as a case study. *Am. Nat.* 160, 147–157 (2002).

Reviewer #2 (Remarks to the Author):

Review of 'radiation continuum'

This is an interesting paper that should find an audience among evolutionary biologists interested in adaptive radiation. The overarching result is unsurprising ("all four major types of macroevolutionary outcomes are represented"), and some potentially major findings ("Less than half of frog families resembled adaptive radiations") are contingent on the subjective identification of both units for analysis and other elements, including the definition of adaptive radiation itself. The authors are not unaware of these subjective factors—these are problems that characterize many comparative studies and most studies of diversification and adaptive radiation—and the authors take steps to attempt to address many of them, performing multiple analyses with parameters tweaked. But their conclusions are sensitive to these perturbations, and there is only so much potential variation that can be accounted for before the number of necessarily arbitrary decisions renders even broadstroke inferences shaky. That all being said, the characterization of a "radiation space" is a strong contribution.

Here are some of the subjective/problematic issues of this paper: taxonomic families are not suitable units for analyses; use of morphological characterization of clades from a tiny percentage of sampled species is imprudent for 'phenotypic diversification' analyses; speciation/diversification analysis of clades from which a small percentage of species is included is even less prudent than phenotypic analyses that undersample phenotypic diversity; adaptive and nonadaptive radiations/nonradiations occur along a continuum so any classification of a lineage or group of lineages as an adaptive radiation (AR) necessarily is the result of arbitrary cutoffs regarding these terms.

Response: We thank the reviewer for their comments, and we address each of them in turn below. The one that is not repeated by the reviewer below is the one about “morphological characterization from a tiny percentage of sampled species” being “imprudent for phenotypic diversification analyses.” First, we morphologically characterized (on average) 25% of the species in each clade. We think that 25% is not necessarily tiny. Regardless, we think that the most relevant issue is whether limited taxon sampling impacts our estimates of phenotypic rates. This issue was addressed specifically for frog ecomorphology by Moen et al. (2021; *Evolution*). Using simulations, they found that sampling only 80 species from a tree of 3,449 species (i.e., 2.3%) did not bias the estimated rates. In response to this important comment, we now point out this previous result explicitly in the Methods, in the section titled “Quantifying morphological diversity and rates of evolution”. We also note that previous analyses have shown that frog morphology is determined largely by microhabitat ecomorphs (e.g., Moen et al. 2016; *Syst. Biol.*), which we have sampled thoroughly here. Furthermore, we have sampled most frog families. Therefore, it is unclear to us how simply including more species of these same ecomorphs and families would change our conclusions.

All of the above issues have stock solutions in print (e.g., for undersampling of species for species diversification analyses: simulation studies, where undersampling of a known distribution results in a reasonable estimate of that known distribution [surprise!], is interpreted as informative regarding reality)—and, again, the authors are not incognizant of these problems. But unfortunately some of these issues are, in my view, insoluble (subjective cutoffs to reach AR) and others simply must await more thorough sampling before they can be applied informatively on this scale (i.e., the scale of all frogs).

Response: Our main diversification analyses used the stem-group method-of-moments estimator, which uses the age and the total number of species in each clade, not the number in the tree. Thus, these results are independent of how many species are sampled in each clade. Therefore, sampling a limited number of species in each clade has no impact on this estimator. Nevertheless, it yields strong relationships between the true and estimated rates in simulations, which we cited in our methods (lines 537–542). We also show how a diversification metric that does depend on the number of species sampled (birth-death estimators) gave very similar results. Therefore we think that the problem of species sampling for estimating diversification rates is not insoluble at all.

The reviewer dismisses the analyses of adaptive radiation because they depend on a subjective cut-off. However, we show that our main results are robust to different cut-offs

(mean vs. median rates) and clade designations (families vs. time-based clades). Specifically, we show that the majority of clades fall into the adaptive-radiation-like quadrant (even if the exact magnitude of that majority can vary). Rather than merely being cognizant of these potential problems, we show (with data analyses) that our results are robust to them.

More on some of these points below:

--Families are not appropriate units of comparative evolutionary analysis, for two reasons: 1) they are arbitrary units, 2) they are not independent.

The authors are aware of the arbitrariness issue and attempt steps to counter this concern by comparing clades of equal age. However, implementation of this admirable safeguard seems to have a major effect on conclusions. The use of families (in my view, the least desirable of clade samples that were used—why would we consider human designations in analyses of the natural world?) returns the '75% of diversity found in 50% of AR clades' result that is reported in the abstract as a (the?) main result—numbers that are used to argue that adaptive radiations "... clearly do not explain nearly all diversity, as often posited." But use of a 120 mya cutoff for clades results in 96/98% diversity occurring in "adaptive-radiation like clades," which would certainly seem consistent with the idea that adaptive radiations explain "nearly all diversity." Which of these, or other possibilities, is the "correct" answer depends on an arbitrary decision regarding the units of analysis.

Response: As an alternative to using families, we examined three different age cut-offs for clades: 80 Myr, 100 Myr, and 120 Myr. We do explain how these results differed, and we explicitly explained why the 120 Myr results differed more than other clade ages. Nonetheless, in response to this comment, we have now added more details of the sensitivity of our results to other methodological choices (lines 212–218).

Overall, these analyses show that most species richness and most morphological diversity is contained in the adaptive-radiation-like clades. The fact that the exact numbers change somewhat based on the exact details of the analyses seems perfectly reasonable and expected, not a unique or fatal flaw. We would expect this pattern with any group of organisms and with other approaches.

Regarding point 2), I may have missed it, but it does not appear to me that any phylogenetic "correction" was implemented for the interrelationships of the families. I realize it is not standard practice to account for phylogenetic covariance in macroevolutionary comparisons of clades; but it should be—diversification rates, like anything else, may display some phylogenetic inertia that must be accounted for (if not, it undermines the analysis of clades at all). Rhacophoridae and Mantellidae share some 80 million years of evolution separate from Heleophrynidae. Wouldn't this shared history suggest some propensity for shared diversification rates in these two families? If you are going to compare across a group of clades (which itself is problematic), minimally you must account for the nonindependence of clades being compared.

Response: All of our statistical analyses—PCA, correlation, regression—were phylogenetically informed. This is explained in detail in the Methods.

--The major conclusions of this paper are contingent on the structure (statistically, the distribution) of the "radiation continuum" (lines 581-585). The construction of this space is interesting and should be useful for visualizing adaptive radiations. But the details of how this space may be useful for hypothesis testing, or even statements of relative frequency, remain to be worked out. Perhaps the construction/discussion of this space is interesting enough for this paper to be accepted—making AR quantitative as a comparative concept is a big deal. But, the framing of this paper as some "test" of adaptive radiations, when this test rests on arbitrary cutoffs of what constitutes an AR, is not compelling.

Response: Please note that our primary focus here is not on making a quantitative test of AR. We are merely describing some clades as being AR-like (as we describe in the Introduction). We emphasize that most of science rests on arbitrary cutoffs. For example, if we had a statistical test for AR, it would rest on an arbitrary cutoff for significance (regardless of whether we used simulations, *P*-values, or model selection). We think that a general property of statistics (and science) should not be treated as a unique failing of our paper.

The reviewer states that: "the details of how this space may be useful for...statements of relative frequency, remain to be worked out." We think that we have worked out these details and demonstrated how this radiation space is useful and robust.

Perhaps it would help to create a null "radiation space" using simulations where species diversification and morphological evolution are simultaneously modeled using simple (Brownian-motion like) models. This paper's current characterization of this space as quadrants centered around means from empirical data is great for visualization but not so great for assessing the relative frequency of patterns in nature.

Response: We appreciate the reviewer's suggestion. However, simulations cannot assess the relative frequency of patterns in nature because they are not based on nature. Or at best, the simulation parameters can be based on observed data (e.g. mean rates of diversification and morphological evolution from empirical data), but then they should simply replicate what is seen in nature. By contrast, calculating rates from empirical data shows the relative frequency of patterns observed in nature.

We do think that simulations could be used to test whether a clade is an adaptive radiation or not (e.g. Moen et al. 2021; Evolution). But any such tests still rely on an arbitrary cutoff for what is considered sufficient evidence for that clade to be an adaptive radiation.

lines 10-13

"what proportion of clades show macroevolutionary dynamics similar to adaptive radiations?"

Second, what proportion of overall species richness and phenotypic diversity do these adaptive radiation-like clades contain?"

These questions are inescapably connected to arbitrary decisions regarding what constitutes an "adaptive radiation" (AR)—that is, how rapidly must speciation occur before it is rapid enough to potentially qualify as an AR. For example: couldn't the results of this paper be spun as new evidence that our definition of AR needs revision, rather than as evidence that ARs do or don't explain diversity as expected? There are no hypotheses to test here; just patterns to report (not that that disqualifies the work scientifically).

Response: Unfortunately, any statistical test for AR will also depend on an arbitrary cutoff for statistical significance. This will be true regardless of the method of testing or statistical framework. The use of arbitrary cutoffs is a general necessity of quantitative science, not a unique failing of our study. What we do is show that our results are generally robust to different arbitrary cutoffs, which we have done in terms of using mean vs. median rates for quadrant boundaries, different diversification metrics, and different ways of defining clades. Specifically, we show consistently that the majority of species diversity and phenotypic diversity is contained in clades that are adaptive-radiation-like.

Lines 132-134

" However, limiting the term "adaptive radiation" only to clades that show significantly elevated rates would mean that most clades have unexceptional rates of diversification and morphological evolution."

This statement epitomizes the issues with subjectivity/perception for these results. What if the real evolutionary pattern in nature is that "most clades have unexceptional rates of diversification and morphological evolution." The characterization of adaptive radiation in this paper apparently disallows this result.

Response: By definition, most clades *must* have unexceptional rates of diversification and morphological evolution; otherwise the term "exceptional" would be meaningless. Only a very few clades with exceptional rates can be considered exceptional (i.e., just as not every child can be above average). This is logic, not some fatal flaw of our paper. As is explained in the paragraph that this short quote is taken from, we need to allow for a continuum between clades with truly exceptional rates and those that are merely higher than average. Moreover, one of our main empirical results (Fig. 4) is that many groups have relatively high rates of diversification and morphological evolution (i.e., they are adaptive-radiation like).

lines 953-55

Aren't the 11 omitted families critical datapoints? I.e., cases where near-maximally low diversification rates occur? (also, it appears that Ascaphidae, which includes only two species,

was included in analysis. Were the 11 excluded families monotypic?).

Response: We agree that omitted taxa are a valid concern in any comparative study. However, there are several key things to note about our data set. First, while we were forced to omit some species-depauperate families, we were also able include many other species-depauperate families with a reasonable proportion of species sampled in each (e.g., Ascaphidae [2/2], Pelodytidae [2/4], Sooglossidae [3/4], Heleophrynidae [2/6], Conrauidae [2/7], Scaphiopodidae [4/7], Bombinatoridae [6/10]). Second, among the 11 families we omitted, Rhinophrynidae is the only monotypic family, whereas the others are mostly species-poor and specimens for them are rare in U.S. natural history collections. This is highlighted most notably by Nasikabatrachidae, which are not found in any institutions outside their native India. Third, four of these 11 families (Ceuthomantidae, Hemisotidae, Nasikabatrachidae, and Odontobatrachidae) are each represented by only a single species in the phylogeny we used (Jetz & Pyron 2018) and are no better represented in any other large, comprehensive anuran phylogeny. Thus, while we would ideally include more of these missing families, we still could not calculate rates of morphological evolution because of their singleton status in the phylogeny. We also know that these clades contribute little to overall species diversity, so their exclusion has little impact on calculating the contribution of clades to overall species richness. Furthermore, since these clades have relatively low diversification rates (as the reviewer acknowledges), then their exclusion does not impact our main conclusion: that most species diversity and phenotypic diversity is contained in adaptive-radiation-like clades. Finally, including these 11 clades would only further support our finding that there are many clades that are not adaptive-radiation like (even though adaptive-radiation-like clades account for most diversity).

Some summary comments:

Methods

The methods used are standard for the field, i.e., they have been utilized in studies published in prestigious journals. In my view, there are insurmountable issues for some of these approaches—in particular, the practices of using families as units of evolutionary analysis and of measuring diversification rate in severely species-undersampled clades should both be retired.

Response: We have demonstrated that these are not insurmountable issues. As noted above, estimating diversification rates (as done in our main analyses) is completely unaffected by taxon sampling within the clade. Furthermore, we showed that most of our results based on families were supported by analyses using age-specified clades (and we discussed the deviations).

Importance

The characterization and visualization of a "radiation space" seems a major contribution. The

conclusions regarding the frequency of "adaptive radiation" are not compelling.

Response: We agree with the reviewer about the latter point, since we are not testing the frequency of adaptive radiation, as was explained in the Introduction. Rather than considering it a main result, we only state the frequency to contrast the amount of diversity this type of evolutionary dynamic explained.

Writing

The writing is mostly clear and easy to follow. There are some general statements that come off as straw-man ("Most of life's vast diversity of species and phenotypes is often attributed to adaptive radiation").

Response: We cited multiple papers that explicitly advocated this statement. Furthermore, we found that most of frog species diversity and phenotypic diversity was contained in adaptive-radiation-like clades. Thus, it is a statement that we support rather than refute. A straw-man argument, by contrast, is poorly justified (which our citations suggest otherwise) and easily refuted (which we do not do).

Figures

Good, effective.

Reviewer #3 (Remarks to the Author):

This manuscript quantifies morphological diversity, morphological rates of evolution, species diversity, and net diversification rates across frog families to define and delimit a radiation space. This novel radiation continuum method is then used 1) to identify the proportion of anuran clades exhibiting macroevolutionary dynamics indicative of adaptive radiations with high rates of both morphological evolution and net diversification and 2) to quantify the proportion of total frog species richness and phenotypic diversity contained within these adaptive radiation-like clades. The authors demonstrate that less than half of frog families fall within the "adaptive-radiation-like" quadrant, yet these lineages represent most diversity. The results presented here will be of broad interest to biologists, and I think this framework will have an important impact on the study of adaptive radiations and quantifying macroevolutionary patterns. I commend the authors for their robust statistical approaches, which included multiple methods and parameter sensitivity analyses for quantifying morphology, diversification rates, and their radiation space. All code and data are available and well organized. I have a few questions and comments for the authors.

My only concern related to analyses is whether the method used to calculate morphological rates (compare.evol.rates) is sensitive to variability in number of species across groups? I generally find the patterns identified in the radiation space for sampled families intuitive as a frog biologist, but I am surprised by some of the outlier families with high rates of morphological

evolution. Most of these outlier families have low species richness and are therefore represented by only a few species in the dataset (e.g., Cycloramphidae, 5 species; Pelodytidae, 2 species; Petropedetidae, 5 species; Alytidae, 5 species). It seems unlikely these families have higher morphological rates of evolution than Bufonidae or Microhylidae, for example. The families with the lowest rates of morphological evolution are also represented by only a few species, but this pattern is less surprising.

Response: We agree that some of these low-diversity families with high morphological rates are surprising. However, many of these families are hardly homogeneous. For example, in Cycloramphidae, compare *Thoropa* and *Zachaeneus*. Similarly, in Alytidae, *Alytes* and *Discoglossus* differ considerably due to occupying different microhabitats. We agree that Pelodytidae and Petropedetidae are more surprising.

On the other hand, we think that Bufonidae and Microhylidae contain large numbers of closely related species with similar morphologies (e.g. think of the hundreds of species that used to be in the genus *Bufo* and the similarity between *Gastrophryne* and *Microhyla*). Even though both families contain some highly divergent species, that is not necessarily the case for most species.

Overall, as we noted in our responses to both Reviewers 1 and 2, the method that was used to estimate multivariate phenotypic rates here was shown by Moen et al. (2021) to be unbiased by limited taxon sampling (using simulations). Certainly, the exact rate estimate will depend on which species are sampled, but low sampling does not bias our rate estimates upward or downward. Diversification rates (as estimated here) should be even more robust to incomplete taxon sampling.

Would it be possible to compare the family-level radiation space to the clade age radiation space by mapping the adaptive radiation, non-adaptive radiation, adaptive non-radiation, and non-adaptive non-radiation quadrants for each approach as traits to the tips of your phylogeny? I recognize that the adaptive-radiation quadrant still included most species diversity and morphological diversity using the clade age approach, but I am wondering how the species composition of the four quadrants have shifted (i.e., are the species within Ceratophryidae still within the NANR quadrant for the 80 my clade age analysis?).

I think it would generally be interesting to visualize the phylogenetic distribution of the radiation space quadrants to assess, for example, whether adaptive radiation-like families are clustered or dispersed across the frog tree of life.

Response: We think that this is a great idea. As noted above in response to Reviewer 1, we have added the radiation quadrants for each family as tip states in Fig. 3. We hope that this will help visualize the distribution of the quadrants across the tree, as noted by the reviewer here. Moreover, we have added an analysis (using the D-statistic) that explicitly tests for phylogenetic clustering or overdispersion. Given the D-statistic is for binary traits, we apply this analysis to each quadrant type (e.g., whether adaptive radiations—versus all other types—are clustered). This test showed that the different radiation types are neither significantly clustered nor overdispersed.

Reviewers' Comments:

Reviewer #1:

Remarks to the Author:

I am glad to see that the authors have done meticulous work responding to my suggestions and those of other reviewers. The manuscript as it stands has addressed all my previous concerns. I do not see any reason to send it out for further review.

Reviewer #3:

Remarks to the Author:

The authors have satisfactorily addressed my previous comments and the revised manuscript has improved. The framework presented here is an important contribution to our field.

My only comment is that there seems to be a plotting error in the revised Fig. 3 for the diversification rate and morphological rate color gradient scale bars (shapes plotted instead of negative values).

NCOMMS-23-07893A: Response to Reviewers

Reviewer #1 (Remarks to the Author):

I am glad to see that the authors have done meticulous work responding to my suggestions and those of other reviewers.

The manuscript as it stands has addressed all my previous concerns. I do not see any reason to send it out for further review.

Response: We thank the reviewer for their constructive criticism in their previous review, which we feel helped improve our paper.

Reviewer #3 (Remarks to the Author):

The authors have satisfactorily addressed my previous comments and the revised manuscript has improved. The framework presented here is an important contribution to our field.

My only comment is that there seems to be a plotting error in the revised Fig. 3 for the diversification rate and morphological rate color gradient scale bars (shapes plotted instead of negative values).

Response: We thank the reviewer for their continued efforts to help improve our paper. We have fixed the plotting error they pointed out.